# Consistency-Checking 3D Geological Models

Marion N. Parquer[1], Eric A. de Kemp[1], Boyan Brodaric[1] and Michael, J. Hillier[1]

Correspondance: Eric A. de Kemp
eric.dekemp@canada.ca
orcid.org/0000-0003-0347-5792

[1]Geological Survey of Canada,
 Three-dimensional Earth Imaging and Modelling Lab
 601 Booth Street, Ottawa, Canada, K0E 1E9

**Abstract**

3D geological modelling algorithms can generate multiple models that fit various mathematical and geometrical constraints. The results, however, are often meaningless to geological experts if the models do not respect accepted geological principles. This is problematic as use of the models is expected for various downstream purposes, such as hazard risk assessment, flow characterization, reservoir estimation, geological storage, or mineral and energy exploration. Verification of the geological reasonableness of such models therefore is important: if implausible models can be identified and eliminated, it will save countless hours as well as computational and human resources.

To begin assessing geological reasonableness, we develop a framework for consistency-checking with geological knowledge, and test it with a proof-of-concept tool. The framework consists of a space of consistent and inconsistent geological situations that can hold between a pair of geological objects, and the tool assesses a model's geological relations against the space to identify (in)consistent situations. The tool is successfully applied to several case studies as a first promising step toward automated assessment of geological reasonableness.

**Keywords** – geological knowledge, geological consistency, 3D geological modelling, temporal relation, spatial relation, polarity

**1 Introduction**

Geomodelling techniques are often deployed to bridge the spatial gaps between explored areas, including gaps in stratigraphic structure, property distribution, and target extent. Increased data availability and rising societal need for natural resources have recently stimulated development of advanced geomodelling modelling techniques such as stochastic simulation (Lajevardi and Deutsch, 2015), time-varying modelling (Hinojosa, 1993), Bayesian techniques (de la Varga and Wellmann, 2016), and direct perturbation of models or data (Lindsay et al., 2012). Wrapped into growing complex workflows (de Kemp, 2016), these new techniques can operate with scarce and heterogeneous data, are frequently deployed to model less accessible and more complex terrains, and often produce a wide range of possible models and associated uncertainties (Wellmann and Caumon, 2018).

However, several problems can arise from these advanced techniques. For example, accuracy issues associated with

scarce data can occasionally become magnified and lead to geologically questionable spatial interpolations, such as

older geological units deposited on younger units (Figure 1). These issues might be further compounded by

decreases in the reliability of the data, as the number of participants increases, or by biases at each modelling step

(Bond, 2015). Data may also become irrelevant due to scale discrepancies, or degraded due to re-sampling to meet

coarser scale requirements or to suit algorithms that imprecisely fit data (Hillier et al., 2021). This can result in

various artifacts such as the well-known implicit interpolator 'bubble' effect (Frank, 2006; Hillier et al., 2016, 2021;

von Harten et al., 2021; Pizzella et al., 2022). As data scarcity and data loss necessarily impact the accuracy and

credibility of any model, multiple realizations are often generated in the hope that some model, or the mean of

models, comes closer to representing reality and minimizing uncertainty. Many simulations also generate model

suites, such as when no priors exist, or when run with the same data or even randomly perturbed data. All these

models, however, are not necessarily geologically possible (Deutsch, 2018). Indeed, some of the more data-driven

3D modelling methods can generate results that respect the data, but do not necessarily respect established

geological principles (Lyell 1833). Conversely, purely knowledge-driven 3D modelling methods might respect

geological principles, or 'norms', but might not fit the underlying data (Bai et al., 2017). Thus, amongst a multitude

of possible models, it is unavoidable that a non-negligible number of them might produce geologically unreasonable

results. This is especially a challenge for hypothesis testing, e.g. climate change scenarios, simulated natural

systems, or various AI training sets, which might involve billions of such models.

The highest quality selection from all possible models then must be achieved, or the geological reasonableness of a

single model must be assessed. This can be accomplished via some combination of  (1) building geologically better

models, or (2) excluding inappropriate models. The first solution involves acquiring more and better data,

knowledge, or algorithms. Increasing the amount of data, possibly from geophysical or structural measurements

(Giraud et al., 2020, 2024; Wellman and Caumon, 2019; Hillier et al., 2014; Grose et al., 2019; de la Varga et al.,

2019), or improving data quality, increases overall accuracy and reduces the number of possible models. Similar

results also might be achieved with increased knowledge, such as input stratigraphy or augmentation of algorithms

with implicit and rule-based approaches (Schaaf et al. 2021, Bertoncello et al., 2013; Bai et al., 2017).

Problematically, however, these solutions typically require the acquisition of new data or knowledge, which is often

impossible. It also might require the development of more geologically robust algorithms to improve model quality

1    (Jessell et al., 2010; Cherpeau et al., 2010; Ranalli, 1980), such as physics-based modelling approaches (Shokouhi et

2    al, 2021; Hobbs et al. 2021), which are not yet mature.

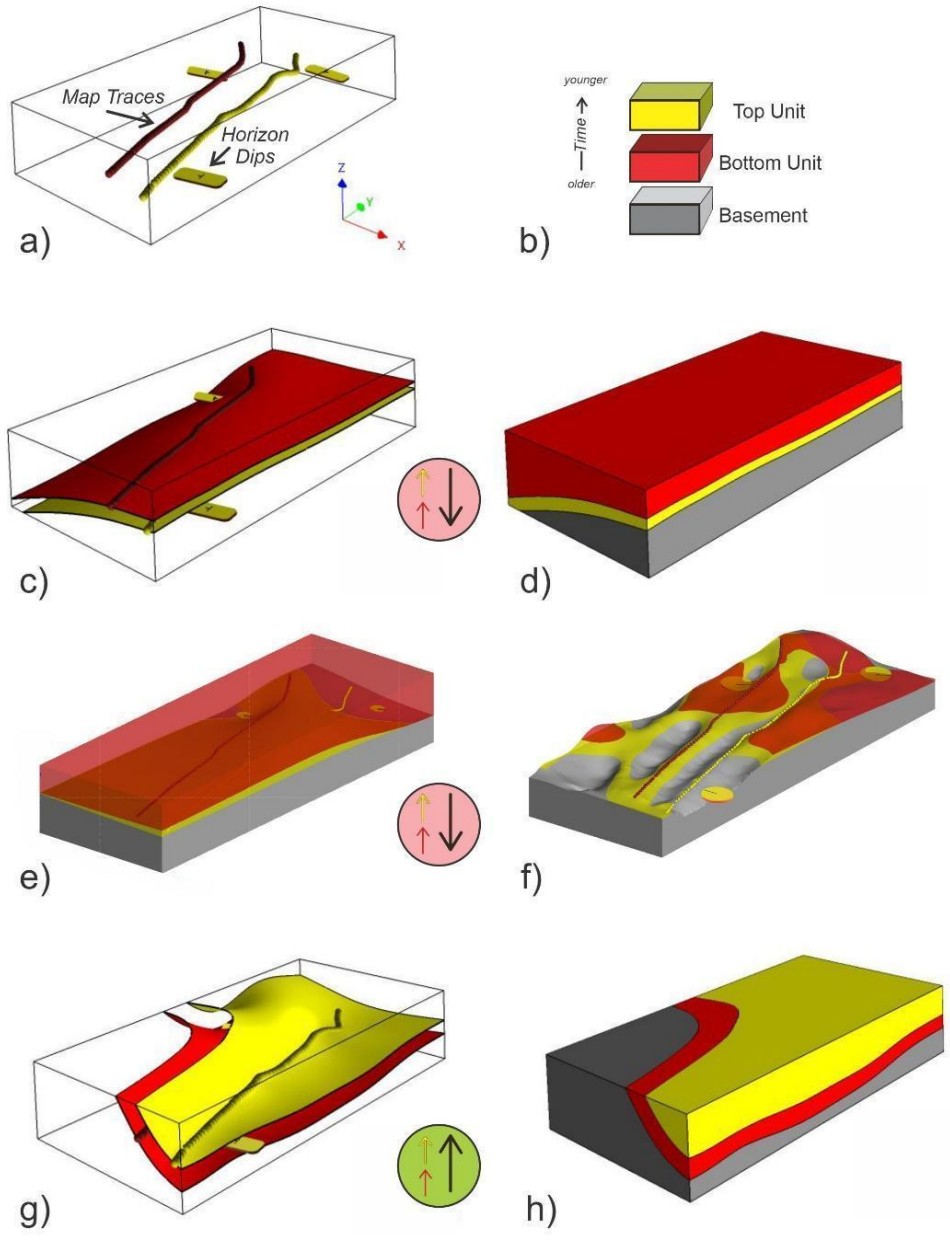

**Figure 1.** Example of *unreasonable* 3D geological models (c) to (f). Sparse input data (a) includes two separate
depositional horizon traces and 3 shallow dipping bedding constraints (yellow-red tablets, yellow is up, red is
down) indicating depositional tops upward. The event history (b) has an older unit (red) deposited below the
younger unit (yellow). However, use of the Lajaunie (1997) implicit method in SURFE (Hillier et al. 2014,
2021) results in the older unit deposited over the younger unit (c)-(d), which is unreasonable in the absence of
other events.  Similarly, using commercial software from LeapFrog Geo (Seequent), in (e) without topography
and in (f) with topography, results in an unreasonable geological sequence with the older unit on top and a
miscalculated geological mapping at surface. In contrast, (g) and (h) show reasonable models generated with
SURFE software tuned to respect a minimal horizon thickness and depositional history. Circled arrows show a

deposition polarity vector for the older unit (red arrow), younger unit (yellow arrow), and the temporal direction (black arrow, from older to younger unit), as well as the geological plausibility of the situation: a green background indicates consistency with geological principles (aligned vectors), and the red background denotes an inconsistent scenario (unaligned vectors).

The second solution, which involves model exclusion, can be accomplished manually or automatically: (i) manually, by having a geologist inspect and reject models using accumulated expertise; or (ii) automatically, by performing a rapid computer-driven check to eliminate poor instances, during or after model construction. A significant disadvantage of the manual approach is lack of reproducibility: as expert knowledge can vary between and within geologists (Brodaric *et al.*, 2004; Brodaric 2012, Bond 2015), it is unlikely manual corrections would be reproducible for more than a few models, and the selection of a certain model would likely be unexplainable. The visualization of complex geomodels is a significant challenge, also making manual validation difficult, time-consuming and likely to miss problems. In contrast, if knowledge is made explicit (Brodaric and Gahegan, 2006), automatic approaches could be reproducible and explainable, as per the consistency-checking approach in this paper. Then, a critical aspect of this approach is the explicit digital encoding of knowledge, as well as its integration into geo-modelling workflows. Although integration techniques like rule-based geomodelling (Pyrcz et al., 2015) and implicit modelling (Jessell et al., 2014) are quite common, they typically incorporate a limited range of knowledge. Extending this range also is not new, e.g. early work focuses on capturing knowledge from a geological map, cross-section, or other field record (Harrap, 2001; Burns, 1975; Burns and Remfry, 1976; Burns et al., 1978; 1969), but only recently have extensions into 3D geo-modelling begun (e.g., Jessell et al., 2021; Rauch, et al., 2019). In addition to limitations in knowledge range, there exist accompanying limitations in its use, as the knowledge is utilized primarily *a priori* for model-building rather than *a posteriori* for model evaluation. Key goals for a consistency checker then include an expansion of the range of knowledge to include an enhanced representation of geological relations, plus an approach for assessing such relations as valid or invalid for effective consistency evaluation.

A first step to such expansion and evaluation might be the utilization of all information from a geologist's observation sheet. However, it is very unusual to incorporate all such knowledge in a 3D model: much of it remains reported on a map, e.g. as colours, abbreviations or symbols, and the rest in the map legend, in related articles and reports, or in the mind of the geologist. In particular, the geological legend as we know can be incomplete (Harrap, 2001) and does not always contain the entire stratigraphic and structural history, prompting the development of a 'legend language' as a first attempt to formalize geological map knowledge and check the consistency of traditional

2D geological maps (Harrap, 2001). Consistency-checking then involves comparison of relations on a map against the 'truth' in a legend; however, legends or other *a prior/*assumed truths, such as stratigraphic columns, might be incomplete, possess errors, or be missing altogether, particularly for under-explored regions such as Mars or many physics-based simulations. Also, it is often difficult to determine if the map or legend is the source of inconsistency. This suggests comparison of a map (or model) against representations of the general rules of geology might be more effective.

Recent investigations into representing general geological knowledge target the topological aspects of geological maps and models (Schafe et al., 2021; Thiele et al., 2016a, b; Le et al., 2013). These focus on the spatial relations between discrete elements of a 3D model, particularly those unchanging under continuous deformation (Crossley, 2005), such as adjacency, inclusion or intersection. An important aspect is the dimensionality of the spatial objects, which might be 0D (a point), 1D (a line), 2D (a surface), or 3D (a volume). These spatial relations are needed for computer encoding to ensure possible object interactions are consistent with, for example, real world physics. Spatial relations between such objects have been widely examined, with distinct relations identified between 2D regions (Egenhofer and Franzosa, 1991) as well as 0D, 1D, 2D, and 3D regions (Zlatanova et al., 2004). They also have been applied to material geological objects (Schetselaar and de Kemp, 2006), providing a basis for the spatial component of geological knowledge, and underpin efforts in knowledge-driven 3D geological model construction (Zhan et al., 2019; 2022). However, they are not yet applied to the evaluation of geological models, especially in combination with temporal relations, despite being applied to the evaluation of models in other domains (e.g. Van Oosterom, 1997; Gong and Mu, 2000; Arora et al., 2021; Nikoohemat et al., 2021; Bezhanishvili et al. 2022).

In this paper we develop a general framework for consistency-checking 3D geological models, a proof-of-concept consistency-checking tool, and test a portion of the framework using the tool in four case studies. The framework consists of a hyperspace of all possible (in)consistent geological relations holding between nine kinds of geological objects, with each relation being a unique combination of a spatial, temporal and polarity relation. The proof-of-concept tool then assesses the relations in the case-studies against a subspace involving four kinds of objects - i.e. depositional and intrusion units, and fault and erosional surfaces - to successfully identify (in)consistencies. Although testing of the full hyperspace, involving all nine object types, is left to future work, the overall framework seems promising and performs as expected on the case studies. The framework is presented in Section 2, the tool is

described in Section 3, the four case studies are presented in Section 4, some additional thoughts on consistency-checking and geological reasonableness are presented in Section 5, and the paper concludes with a brief recap in section 6.

**2 Geological Consistency-Checking Framework**

Geological data and knowledge have been accumulated over thousands of years of human inquiry into our natural environment, with modern formal geological knowledge emerging in the mid 1800's (Lyell 1833; Rothery, 2016). A collective understanding is found in digitally archived articles and books (e.g. Kardel and Maquet, 2012), in online products and courses (e.g. Fattah, 2018), and in several formal ontological articulations (Brodaric and Richard 2021, Garcia et al. 2020, Perrin et al. 2011, Brodaric, B. and Gahegan 2006). It is particularly useful to help understand the often hidden and unobserved subsurface of the Earth. However, the various possible sources of data (e.g. surface mapping, boreholes, geophysical surveys) generally cannot provide sufficiently uniform and continuous information for a volume of interest. Supplementary geological knowledge is required for improved interpretation between sometimes extremely scarce observations (Groshong, 2006; Frodeman, 1995), especially when coupled with new data integration techniques and approaches (Giraud et al. 2020; Wellmann and Caumon, 2018).

For consistency-checking purposes herein, we distinguish between data and knowledge, with data being observational, and geological knowledge being either local or universal. Data then includes any form of observation used to understand a specific geological situation, e.g. bedding top indicators, structural orientations, fault and horizon contacts, seismic picks, or other geophysical readings. Local knowledge applies to a specific area but is not observational: it is interpretational and includes things such as the local stratigraphy and process history. In contrast, universal geological knowledge is applicable to different geographical areas and includes things such as general laws, principles, process types, and classification systems, e.g., Walther's Law, uniformitarianism, the notion of deposition, rock type classification. Significantly, data and knowledge are interconnected insofar as knowledge is inferred from data, and the data is contextualized by knowledge during observation and interpretation (Brodaric et al, 2004). Indeed, both data and knowledge are required to arrive at any interpretation, including a 3D geo-model. Consistency then can be seen as the degree of agreement between a model and the relevant data and knowledge. However, current modelling techniques are primarily focused on ensuring and assessing data consistency, with knowledge consistency less developed, e.g. implicit modeling techniques typically optimize fit to data and assume

stratigraphic consistency, but such consistency might not be achieved by all techniques (see Figure 1), and further might not be reflected in all geometric realizations due to idiosyncrasies of spatialization algorithms (Hillier et al., 2021). Therefore, some output geological models can still fail to respect basic geological principles.

To determine knowledge consistency for a 3D geo-model, we expect local knowledge to be typically derived from a 2D map legend, cross-section, or associated report, with the geological processes and the combined event histories being discerned through geologically possible binary relations. For example, the contact relation between two adjacent depositional units can be decomposed into a spatial relation (spatially touching), a temporal relation (temporally adjacent), and polarity relations (aligned material gain or loss), and each of these can be evaluated separately for consistency with established geological knowledge.

CC Truth Tables, or consistency checking truth tables, then denote all possible combinations of these relations for pairs of object types, with each combination identified as (in)consistent. Knowledge consistency is finally assessed by traversing the spatial relations between pairs of objects in a geo-model, using the local knowledge to determine object types, temporal relations, and polarities of the objects, which together form an index into the truth table, which denotes universal knowledge, to determine the (in)consistency of a specific relation.

**2.1 Geological Objects and Polarity**

The geological objects in a 3D geo-model (geo-objects) are, for the purposes of this paper, representations of instances of nine distinct geological object types: depositional unit, intrusion unit, extrusion unit, metamorphic unit, fault, erosion surface, fold volume, and linear and planar fabric. This list is not comprehensive, but reflects an initial suite of key entity types found in models.

Each geo-object is either material or immaterial. A material geo-object is constituted by some rock material and is volumetric as it occupies 3D space. An immaterial geo-object is not constituted by any rock material, but (1) might be volumetric and occupy 3D space, such as a fold which occupies the space of its host rock, or (2) is not volumetric and occupies lower-dimensional space, such as a 2D fault or erosional surface. Note that horizons, understood as the top or bottom surfaces of a volume, are excluded from the geological object types primarily because, in effect, they imply a volume and are thus already incorporated into the volumetric types. This does not exclude the top or

1    bottom surfaces of material entities from being represented in 3D geo-models, but they are not distinct geological

2    object types in this paper and are converted to 3D volumes for consistency-checking in our proof-of-concept tool.

Additionally, we utilize two types of polarity associated with geological objects: internal polarity and temporal

polarity. Internal polarity is a vector within a geo-object roughly pointing in the direction of creation or destruction

of the object's material, or in the growth direction of the object's boundary: e.g. for depositional units, from the base

or oldest part of the geological body to the top (in the direction of material accumulation), for erosional surfaces

from the top to bottom of the eroded rock body (in the direction of material destruction), and for igneous units from

the core to the distal geological contacts with host rocks (in the direction of boundary change). Although material

geo-objects generally possess a global internal polarity, some immaterial geo-objects of lower-dimensionality lack

polarity as they are not associated with material growth or destruction, e.g. fault surfaces, while other immaterial

geo-objects, such as an erosion surface, possess an internal polarity pointing in the direction of material destruction

of the eroded unit.

Geo-objects also might have many local internal polarities distributed throughout the object, constituting an internal

polarity field and forming the basis for determining its global polarity. Significantly, although we strictly use global

polarity in this paper, the overall framework developed herein does not depend on it and would equally function

with local polarities. Note there are pros and cons associated with each type of polarity. Although data for global

polarity is generally more available and easier to implement in tools, it could be hard to estimate in certain

situations, e.g. radial cooling directions for intrusions in which a single vector trend does not suffice. In contrast,

local polarities are often difficult to obtain and harder to implement in automated tools.

Temporal polarity is an age direction vector that represents an oriented age relation held by two geo-objects,

pointing from the older to the younger object and set parallel to one of the object's internal polarity. As a vector, and

in contrast to a typical relation, it orients the age relation in space, thus enabling comparison with internal polarity

vectors as well as direction-oriented space-time analysis of geo-object interactions.  Collectively, there can exist

three polarity vectors associated with a pair of geo-objects: the internal polarity of each object and the temporal

polarity holding across the objects. The alignment of these vectors then helps determine the geological plausibility

of the situation (see Figure 1). The nine types of geo-objects, and associated polarities, include:

- Depositional unit: a material rock volume formed primarily by processes like gravity, water, or air transporting and accumulating materials over a specific time interval. The internal growth direction of this unit is mainly vertical and points upward, from the bottom to the top of the unit, opposite to the force of gravity at the time of deposition (Figure 2a). Although these units can extend laterally over a large area, their formation is driven by near-vertical deposition.

- Extrusion unit: a material rock volume primarily generated by igneous extrusive processes and associated with a time interval. The local internal polarities typically point radially upwards to a proximal vent or feeder facies. This includes internal polarities associated with deposition of eruptive material, which is affected by gravity and tends to flow downhill, but with airfall material accumulating upward. A global internal polarity vector thus points upwards at the time of formation, similar to sedimentary units. However, extrusive units with variable growth direction, such as in subglacial situations are an exception, having chaotic eruptive depositional internal polarity vectors that cannot be characterized by a single global vector; a global internal polarity vector thus would be absent for such units.

- Intrusion unit: a material rock volume primarily generated by igneous subterranean processes and associated with a time interval. Its internal polarities radiate from a core region towards the cooling host rock contact surfaces (Figure 2c), with a global internal polarity set to a representative direction. This polarity can be seen as boundary growth - the growth direction of the boundary of the unit - often in opposition to material accumulation as intrusions tend to have new material added to their core. Many configurations for the growth gradients in these bodies exist, but in general the emplacement contacts with host rocks are similar to unconformities, in that they tend to be truncating earlier material through magmatic erosion, assimilation or expansion (Annen, 2011).

- Metamorphic unit: a material rock volume primarily generated by deep thermal-kinetic-chemical processes and associated with a time interval. The internal polarities are perpendicular to the metamorphic isograd and point to the lower metamorphic grade or into the host protolith (Figure 2d). In many cases a global internal polarity vector can be set pointing upwards from a core heat source. This holds for a regional perspective, in which we can envision the earth's regional geothermal gradient as pointing from hotter-deeper to cooler-shallower lithospheric material. It also holds for a local perspective, in which the location of the source of metamorphism, and hence the local gradient, may be easier to establish from metamorphic

aureoles around intrusions. The metamorphic unit geo-object is included herein to allow analysis of thermal-kinetic-chemical gradients with respect to other related geological features.

- Fault surface: an immaterial 2D surface between displaced rock volumes that were once continuous, and associated with a time instant or interval for the displacement activity (Figure 2e). The surface lacks internal polarity, as it is never constituted by any material. Fault surfaces are distinguished from fault blocks or zones (Qu et al. 2023), with the latter material and volumetric, but not considered in this paper.

- Erosion surface: an immaterial 2D surface where a rock volume has completely or partly eroded via a mechanical or chemical process. It is associated with a time interval or instant indicating the end of the erosion process. Its global internal polarity points in the direction of material destruction (Figure 2f).

- Fold: the shape of the underlying host rock often caused by various tectonic and/or gravity-driven processes within a time interval (Figure 2g). Because shape is a characteristic (or property) of its host, like colour, size or thickness, it cannot be a material entity, so folds are immaterial. Such characteristics also are not parts of their host: a rock unit's characteristics such as shape, colour, or thickness are not a fragment of the unit.  The host, however, might be either material or immaterial: host rock units are material, but host faults or erosional surfaces are immaterial. As folds occupy the space of their host, they further can be volumetric or lower-dimensional. Herein we consider folds as immaterial objects without internal polarity, but they might have a form of kinematic polarity, such as vergence and tectonic transport direction, which we do not address in this work.

- Linear fabric: a penetrative linear orientation of some rock material with an associated time interval. Specifically, the fabric is a whole with its material parts aggregated in a linear orientation, thus the fabric is material and volumetric. Some linear fabrics could have a unidirectional global internal polarity (Figure 2h), such as from paleocurrents, or a bidirectional global internal polarity, such as from tidal currents.

- Planar fabric:  a penetrative planar orientation of some volumetric rock material parts, with an associated time interval. A primary planar deposition fabric has a positive upward polarity at the time of formation (Figure 2i), (i.e., bedding top observations). A metamorphic planar fabric in general has no polarity. Igneous fabrics might have an internal polarity direction from crystal accumulation, igneous flow layering, or emplacement contact directions. Fabrics in general are key to resolving complex event histories (Burns 1988). As all fabrics are composed of materials arranged in a certain spatial orientation, and these materials are part of a host rock unit, then fabrics are also a material part of their host. This differentiates fabrics from

folds in this paper: in contrast to fabrics, folds are composed of shapes that are immaterial characteristics, not parts, of their host.

Field geologists typically infer these geo-objects, and associated geological histories, by interpreting repeated geological relations across field sites, suggesting the presence of a simple topological framework underlying variously complex geological situations. Such relations further can be decomposed into combinations of spatial, temporal, or polarity relations. For example, if depositional unit Sandstone-A is *intruded-by* intrusion unit Granite-B, then we also expect a spatial relation to hold such as Sandstone-A *spatially meets* Granite-B, a temporal relation to hold such as Sandstone-A *is temporally met by* Granite-B, and the global internal polarities are either *aligned* or *opposed*. A consistency checker then must verify the validity of such relation combinations.

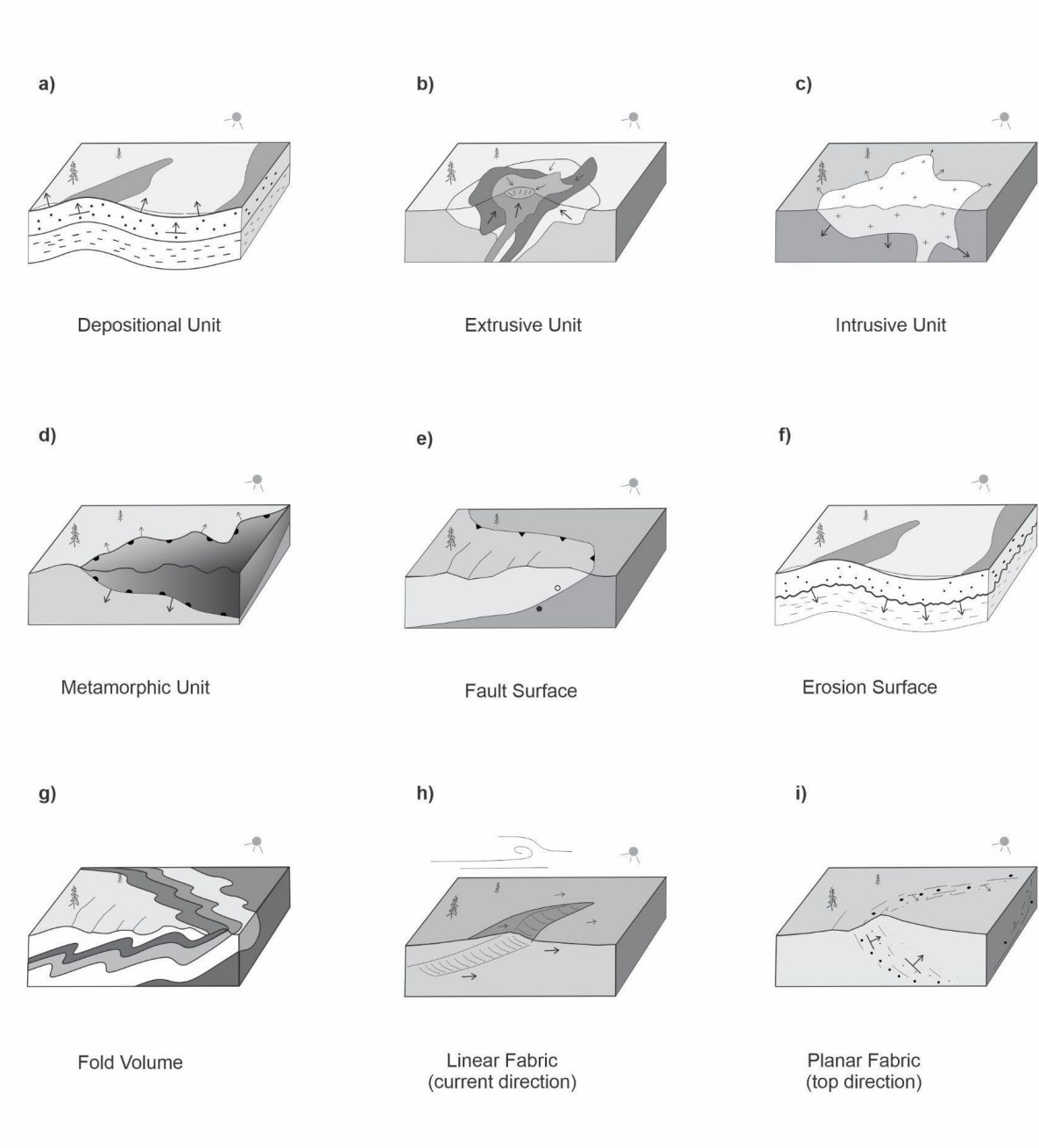

**Figure 2.** Examples of geological objects with polarities, symbolized with black arrows. Metamorphic unit (d) is a contact aureole around an intrusion, isograds ornamented on the warmer side. Note fault features do not have internal polarity (e). Planar depositional point observations depicted in (i).

**2.2 Spatial Relations**

Prominent formalisms for binary spatial relations are derived from two main approaches (Galton 2009), Region

Connection Calculus (RCC) and the 9-intersection model (9I; Egenhofer, 1989; Egenhofer et al., 1993, Egenhofer

and Franzosa, 1991). In this paper we informally adapt the 9I approach , implemented for 0, 1, 2 or 3D objects and

512 possible spatial relations (Zlatanova et al., 2004). However, these 512 possibilities are drastically reduced for

typical geological situations in 2D and 3D (Schetselaar and de Kemp, 2006), resulting in 40 spatial relations for the

nine geological object types, as shown in Figure 2; then for any pair of spatial objects only one spatial relation can

hold. These relations can be represented as a three-part tuple, as shown in Tuple Equation 1. The tuple is also

directed or not, depending on the symmetry of the relation, given that asymmetric relations are directional and

symmetric relations are not directional; e.g. *meets* is symmetric, so if A *meets* B then B meets A, thus *meets* is not

directional; but if A *contains* B then it cannot be the case that B *contains* A (or A *is contained by* B), so *contains* is

asymmetric and directional. The symmetric spatial relations from Table 1 are *is disjoint with*, *meets*, *overlaps*,

*equals*, *intersects*, and the remaining relations are asymmetric. Symmetric relations also are their own converse,

whereas asymmetric relations have distinct converses, such as A *contains* B and B *is contained by* A.

$$Entity_A \left\{ \begin{array}{l} isdisjointwith \\ meets \\ overlaps \\ contains \\ iscontainedby \\ covers \\ iscoveredby \\ equals \\ intersects \end{array} \right\} Entity_B \qquad (1)$$

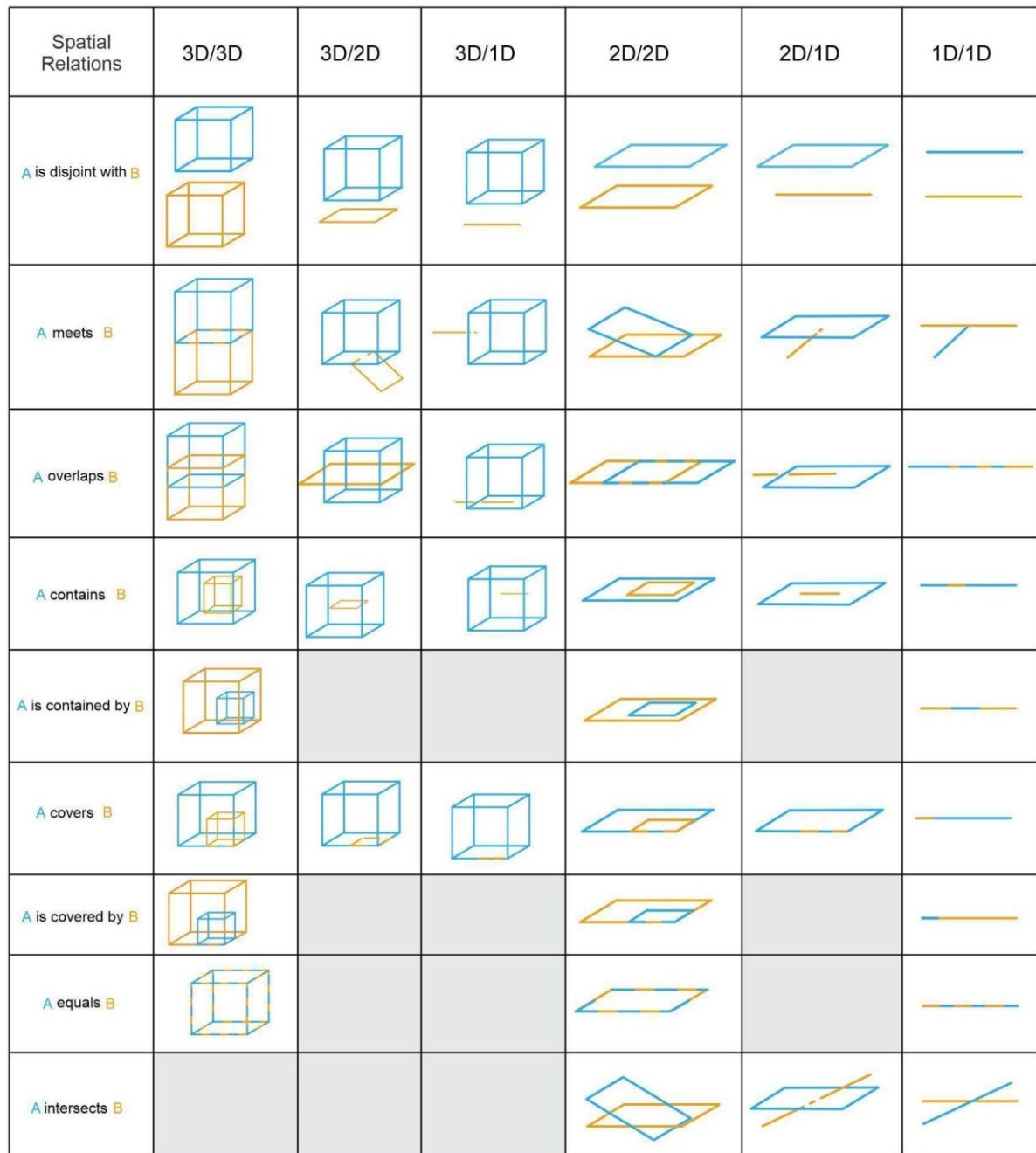

3  **Table 1.** The 9 spatial relations between two geological objects of 1/2/3 dimensions. Blank gray cells denote
4       impossible spatial relations; after Egenhofer (1989), Egenhofer et al. (1993), Egenhofer and Franzosa (1991),
5       and Zlatanova et al. (2004).

**2.3 Temporal Relations**

Temporal relations are required to establish a temporal ordering between geological objects (Perrin et al., 2011). Though the temporal position of a geological object is not always known (Michalak, 2005), the temporal ordering between objects can be derived from the timeline of associated generative events (Galton, 2009; Claramunt and Jiang, 2001). As with spatial relations, dimensionality plays a role: temporal relations can be categorized according to the nature of the time duration (of the event) with 3 potential combinations: period/period, period/instant, or instant/instant. Building on Allen's definitions (Allen, 1983), this leads to 14 distinct temporal relations, including converses (e.g. A *precedes* B and B *is preceded by* A), as shown in Table 2, for the nine geological object types; moreover, for any pair of objects only one temporal relation can hold. Of note is the *is incomparable to* relation, which indicates the temporal ordering is unknown due to unavailable temporal knowledge about one or both objects. Though instantaneous event durations are unlikely in reality, they are common in recorded knowledge and data, thus time instants are valuable to the framework. Tuple Equation 2 illustrates the three-part tuple for expressing these relations. The symmetric relations are *equals* and *is incomparable to*, with the remainder being asymmetric.

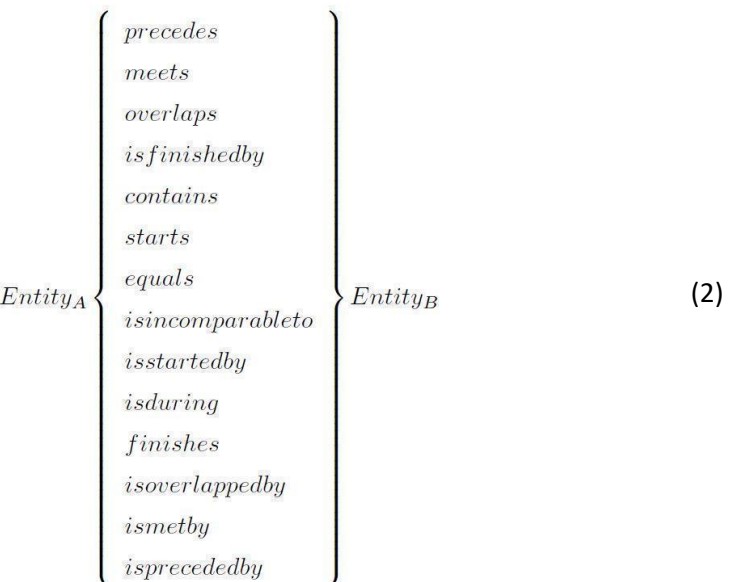

$$Entity_A \begin{Bmatrix} precedes \\ meets \\ overlaps \\ isfinishedby \\ contains \\ starts \\ equals \\ isincomparableto \\ isstartedby \\ isduring \\ finishes \\ isoverlappedby \\ ismetby \\ isprecededby \end{Bmatrix} Entity_B \qquad (2)$$

| temporal relations | period / period | period / instant | instant / instant |
|---|---|---|---|
| A *precedes* B<br>B *is preceded by* A | | | |
| A *meets* B<br>B *is met by* A | | | |
| A *overlaps* B<br>B *is overlapped by* A | | | |
| A *starts* B<br>B *is started by* A | | | |
| A *finishes* B<br>B *is finished by* A | | | |
| A *during* B<br>B *contains* A | | | |
| A *equals* B | | | |
| A *is incomparable to* B | | | |

**Table 2.** The 14 temporal relations between two geological objects, after Allen (1983). The temporal timeline advances from left to right in each cell. Blank gray cells denote impossible temporal relations, and blank white cells denote unknown temporal relations.

### 2.4 Polarity Relations

A polarity relation can be determined from up to three independent component polarities (discussed earlier in Section 2.1): the two internal polarities, dependent on the type of geo-object and its creation processes, and the temporal polarity. The internal and temporal polarity vectors can be compared to determine if they are 'aligned' or 'opposed':

- Aligned polarity relation: the vectors are roughly parallel, such that each vector is within 90° of every other vector.
- Opposed polarity relation: a vector is oriented in an opposite direction to the others, such that one vector is at least greater than 90° from one of the others.

Importantly, polarity alignment or opposition does not necessarily determine (in)consistency alone, as such determination requires consideration of the spatial and temporal relations. For example, opposed internal polarity can indicate either inconsistency or consistency: e.g. depositional units that spatially meet and have opposed internal

or temporal polarities are inconsistent (Figure 3b), because such units must create material in the same spatial and

temporal direction; but a touching depositional unit and erosional surface with opposed internal polarity are

consistent, because the surface must erode material towards the older unit (Figure 3e). The internal polarity relation

also might not play a determining role in assessing (in)consistency, as the spatial and temporal relations may

individually or together be determining factors: e.g. the (in)consistency of an intrusion into a host depositional unit

is determined regardless of internal polarity (Figure 3c-d), as the intrusion must be younger and touching the unit,

otherwise some interceding object such as a fault or erosional surface is missing from the model; similarly for a

depositional unit and a fault surface (Figure 3g-h), as the unit must pre-exist the fault. Note we do not consider

growth faults to be strictly synchronous within a full unit, since at least some of the material needs to be in place

first, prior to faulting. Additional examples of consistency-checking with polarities are shown in Appendix B.

The requirement for this complex polarity relation might not be intuitive, but it is driven by the need for wide

applicability across diverse geological situations and knowledge environments. Immediate simplifications are

limited and do not generalize. For example, checking a model solely against *a priori* local knowledge is not always

possible, due to its incompleteness, incorrectness, or absence, and related sources of inconsistency - model or

knowledge - are often indeterminate. The temporal polarity relation alone also is insufficient: e.g. even in simple

depositional environments, consistency assessment requires knowledge of spatially above and below relations -

younger units are above older units - and these spatial relations are typically hard to determine computationally.

Moreover, such simplifications fail in complex geological situations: e.g. spatially above and below cannot be

determined for a depositional unit pair stacked side-by-side, perhaps due to tectonism; and the temporal vector on its

own cannot discriminate the valid and invalid spatial configurations for this pair, but these can be resolved with the

polarity relation.

A general framework for (in)consistency therefore must take into account the spatial, temporal, and polarity

relations. This is accomplished by using these relations as an index into truth tables representing geological norms

and specifying the (in)consistency of the situation (see Section 2.5).

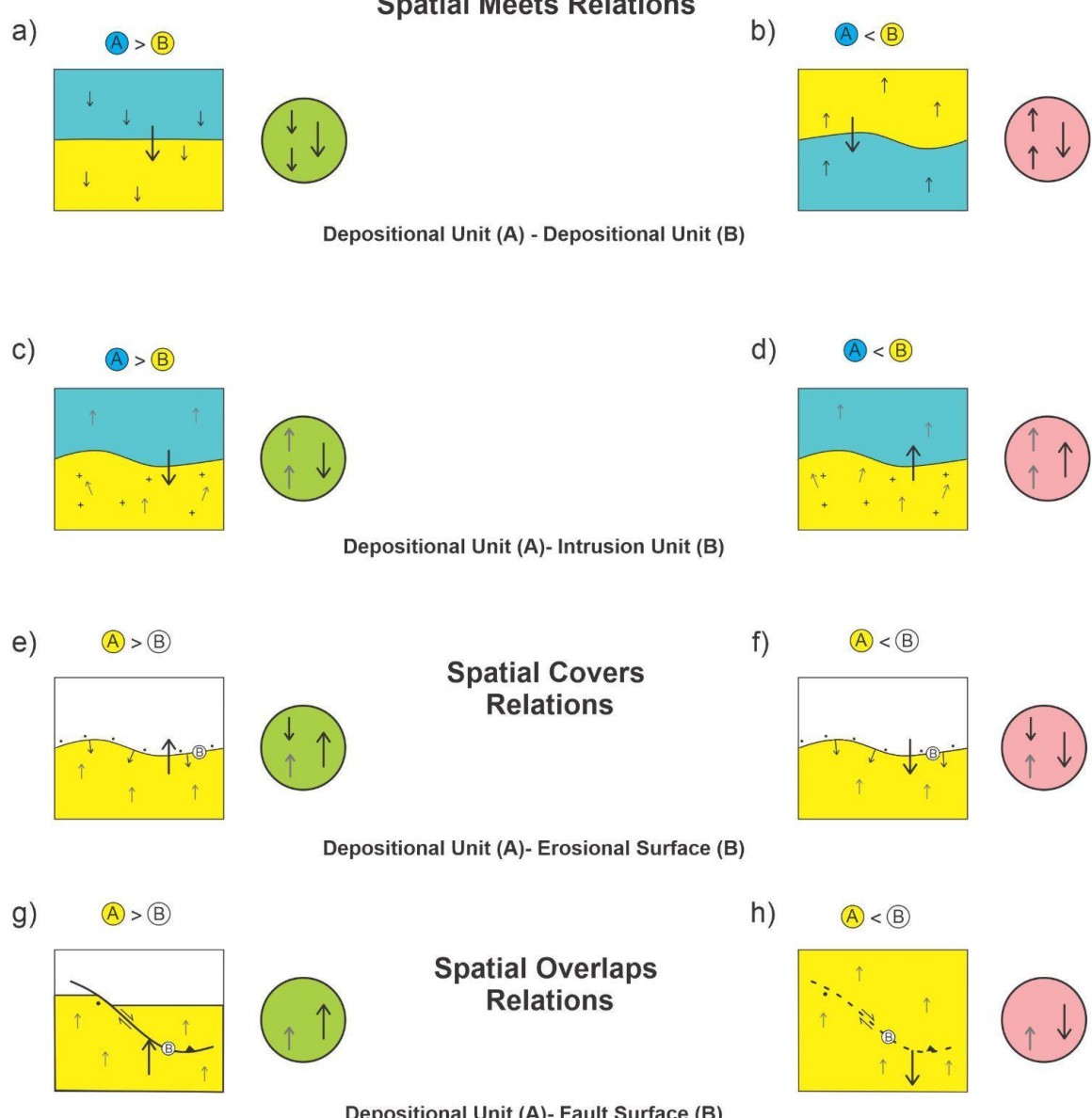

**Figure 3.** Examples of consistent (green circle) and inconsistent (red circle) polarity configurations for the spatial *meets, covers* and *overlaps* relations, and for the temporal *meets* relation, e.g. A > B is A temporally *meets* B, and A < B is A *met by* B; included are two vectors for internal polarity (small arrows) and a third vector for age polarity pointing from older to younger object (large arrow); top small arrow in circle is for A, and bottom is for B. For the two depositional units depicted in (a), the entire package is overturned by a later process, but

nevertheless is consistent. This would still be the case if the package of units is rotated by any angle, including vertically where there would be no sense of relative below and above for the units. An inconsistent scenario between such units is depicted in (b), as the age polarity is opposed to one of the internal polarities, implying reverse deposition of older on younger material. In (c) the deposition - intrusion unit scenario is consistent, as the age vector is aligned with the intrusive process, whereas in (d) it is not allowed because the host depositional unit is younger than the intrusion unit. In both (c) and (d) the internal polarities do not impact consistency evaluation, as age is the determining factor. In (e) and (f) the erosional surface needs a pre-existing material to erode, thus the direction of material reduction of the eroding surface moves into the older underlying depositional material. For a consistent scenario the age vector points from the older unit to the younger erosional surface (e), and for an inconsistent scenario, the age vector points from the older surface to younger unit (f), with only 2 opposing polarity vectors needed to determine validity, age vector and erosion direction. In (g) the fault has no internal polarity but needs a material object to displace. When the depositional unit pre-exists, as in (g), the relation is valid; however, in (h) the depositional unit is younger and did not exist when the fault evolved, so the relation is inconsistent. Grayed arrows indicate the internal polarity vector is not essential for truth table consistency.

## 2.5 Geological Principles

Many geological principles, known implicitly to geologists, must be considered in assessing, even grossly, the

consistency of the spatial, temporal, and polarity relations between two geological objects (Ziggelaar, 2009; Aubry

et al., 1999). Amongst the foremost are the following considerations:

- principle of lateral continuity: in general, a given depositional unit tends to have a similar age over its

  full extent. Diachronous and heterochronous units are not uncommon.

- principle of actualism: past objects are formed by processes (tectonism, magmatism, deposition …)

  acting in the same way as today.

- principle of paleontological identity: two objects with the same association of stratigraphic fossils are

  considered contemporary.

- principle of superposition: without structural disruption events, a given object is younger than the

  object it overlies and older than the one overlying it.

- principle of horizontality: sedimentary objects, have initial nearly horizontal orientation; a non-

  horizontal sedimentary sequence is generally deformed after its deposition with faulting, slumping or

  tectonic folding. Local exceptions occur such as syn-sedimentary deformation.

- principle of cross-cutting: a given material layer is older than objects cross-cutting it.

- principle of inclusion: an object included into another object is older than the including object (clasts in

  a conglomerate or a volcanic flow picking up older material), except when a younger object internally

  displaces the enclosing object (i.e. geode, dyke, sill, migmatite melt phase).

For the nine types of geological objects considered herein, 45 valid pairwise combinations of objects are possible, but this paper focuses on 7 key tables and subspaces relevant to the case studies (see Code and Data Availability). For each object combination, a ternary CC Truth Table establishes all possible consistent and inconsistent spatial-temporal-polarity relations between the geo-object types: spatial relations along one side, temporal relations along another side, and internal polarities along a third side; each table cell then can be marked as consistent or inconsistent for the pair of objects. Alternatively, the truth tables can be seen as denoting a five-dimensional hyperspace representing all possible geological relations, with axes corresponding to two geo-object types and their spatial, temporal, and polarity relations. Values along the axes are the discrete relation types, e.g. the spatial axis has values for *is disjoint with*, *meets*, etc. Consistent values are objects in this space, while inconsistent values occupy empty points in the space. For example, a consistent object might be found at (*depositional unit, depositional unit, spatial meets, temporal meets, aligned*), but the space is empty and inconsistent at (*depositional unit, depositional unit, spatial meets, temporal precedes, aligned*). When polarity is irrelevant, the cell values are the same for both aligned and opposed rows, leaving the polarity subspace empty and set to null. For instance, (*depositional unit, intrusion unit, spatial meets, temporal meets, null*) is consistent when the depositional unit is older than the intrusion unit it touches. Note the polarity axis remains necessary for the other cases in which polarity co-determines consistency.

**Table 3.** CC Truth Table showing consistent (green) and inconsistent (red) spatial-temporal-polarity relations
between two depositional units. All 14 temporal relations are not included (as columns) as the values are
duplicated for the inverse temporal relations.

The CC Truth Table in Table 3 illustrates all possible spatial-temporal-polarity relation combinations for two

depositional units. The eight columns represent the temporal relations possible between two intervals of time; the

remaining inverse temporal relations are excluded for reasons of space and redundancy, as the values in each row

are repeated for the temporal inverse, e.g. A *precedes* B and A *is preceded by* B are both red. The rows in a truth

table represent the possible spatial and internal polarity relations between two depositional rock volumes. Green

cells then indicate consistent combinations, red cells inconsistent combinations, with the consistent cells being far

less numerous. Indeed, in Table 3, two distinct depositional units can be spatially related only via *is disjoint with* or

*meets*, once material sharing is excluded (see assumptions below). All combinations are possible for spatially disjoint units, but only aligned polarity is valid for units that spatially meet, because opposed polarities would signal inconsistencies, such as missing events or intermediary objects. As the truth tables are not necessarily columnar symmetric, the complete tables are provided in the supplementary files (see Code and Data Availability).

In addition to the general geological principles, the following assumptions govern the tables:

- Relata: are the two geo-objects participating in a binary relation, with their type fixed across all relations in a truth table. For example, for a truth table between a depositional unit A and intrusion unit B, A is the first participant and B is the second participant for all relations in the table, e.g. A *meets* B, A *precedes* B, B *is preceded by* A, and A *is aligned with* B. This ensures all possible relation combinations are considered for the pair of objects.

- Time: the framework assumes a geomodel is assessed for consistency at a single point in time. The objects in a geomodel, of course, can develop over different times, but it is their state at a specific time that is evaluated. This impacts the validity of certain geological relations, which might be invalid at a timepoint but valid across timepoints: e.g. two material units cannot share space at a timepoint, but might occupy a common space at different times. There are two main reasons for this choice: (1) practically, most models are developed to reflect a state of geological reality at a single timepoint (typically today); and (2) assessment across time will increase the number of consistencies, and reduce the number of inconsistencies, as many more situations are possible, dramatically increasing complexity and reducing the effectiveness of any consistency-checking approach.

- Space: it is assumed geological objects can be spatially disjoint and possibly very far apart, e.g. on different continents, thus allowing all temporal relations to hold in such cases.

- Space-time: the time assumption implies the objects being assessed are so-called endurants or continuants, which are fully present at a timepoint, i.e. all parts that can be present at a timepoint are present, such as for a rock, geological unit, or fault surface. This contrasts with so-called perdurants or occurrants, e.g. space-time worms (4D spatio-temporal objects), processes, or events, which are not fully present at any timepoint, but unfold in time, so are composed of temporal parts that accumulate over time. Then only a temporal part can be fully present at a timepoint, but never the whole worm, process, or event, unless it is instantaneous. Perdurants/occurrants are spatially located at the position of their endurant/continuant participants, and both

participants and their location can change in time. E.g. a ground shaking event - an earthquake - might have discrete early, middle, and late parts, and have the ground and various buildings as participants, but the whole shaking event is not fully present at any timepoint, because it requires all three parts to be complete. The framework assesses only endurants/continuants, and does not check for correct process behavior, for example in simulations.

- Material sharing: we assume it is physically impossible for macroscopic material objects (endurants/continuants) to share space at a single point in time, unless they share parts, such as one being a part of the other, which restricts the allowable spatial relations between these objects. Consequently, if models are evaluated at a single timepoint, then material sharing is impossible for the material geo-objects outside of part-whole situations, such as a lithology and a geological unit, a formation and a member, or a fabric and its host material unit. We further assume material objects must be volumetric, and can share space with immaterial objects, either volumetric and non-volumetric. E.g. a filled hole shares space with its filling material, and a non-volumetric surface on a material object shares lower-dimensional space with the object. Other non-material objects, such as qualities, e.g. the colour, size, thickness, or shape of an object, also share space with the object carrying them: the grey colour of a rock is not made of material, but occupies the space of its carrying material.  It is also tempting to consider tightly intermixed material objects to share space, but this is a physical impossibility – these are simply objects with mixed composition that share neither space nor material at a timepoint. It is also tempting to consider metamorphic units to share space with other units, typically older, but this too is physically impossible at a timepoint, unless one unit is part of the other. In fact, this metamorphic scenario typically consists of the units sharing space, not material, at different times. However, nothing prevents a user or tool from treating metamorphic units as precursor units, e.g. protoliths, during consistency-checking. Although some immaterial objects, such as holes, might share space at a timepoint, e.g. the pore-space of a formation shares space with the pore-space of its member part, these immaterial parthood situations also are excluded.
- Parthood: although material and immaterial wholes share space with their parts at a timepoint, we exclude such space sharing from the current framework, leaving it to future work. This restricts consistency-checking among certain geo-object pairs, such as between a depositional unit and its material parts, e.g. a group and formation, or the unit and a fabric. However, we do not consider this to be a severe limitation for now: the exclusion does not invalidate the framework nor its use for the very many non-parthood

situations; and spatial parthood situations are not currently output by most modeling algorithms. All prevalent algorithms, that we are aware of, will partition objects into non-overlapping spatial regions by design; so if geometric representations from these algorithms have spatially overlapping regions (including for the metamorphic scenarios), then there exists an inconsistency. In future work, we expect parthood to be an additional dimension in our hyperspace, added to the space, time, polarity, and object type dimensions.

- Model completeness: 3D geo-models are assumed to be complete. Therefore, any two geological objects that touch cannot have objects missing between them, such as an intermediary erosional surface or fault. Without this assumption, the range of consistent scenarios becomes extremely large, with significantly fewer inconsistent scenarios, and the effectiveness of the approach diminishes. Conversely, with this assumption, inconsistent scenarios can signal (but not identify) the absence of spatial intermediaries, which is useful during model-building.

**3 Geological Consistency-Checking Tool**

The consistency checker workflow is presented in Figure 4. This workflow aims to detect the consistency of 3D geo-models given knowledge inputs of:

- a 3D geo-model;
- local knowledge consisting of relative or absolute ages, internal polarities, types of geological objects, and
- universal knowledge in the form of truth tables reflecting geological norms.

After traversal of the 3D geo-model, the consistency checker constructs three intermediary products:

- a geo-object list, itemizing the geometric objects in the geo-model;
- a matrix of temporal relations for each pair of geological objects;
- a matrix of spatial relations for each pair of geological objects;

Then, as per Algorithm 1 (see Appendix A): for each pair of geologic objects, the checker obtains their spatial relation from the spatial relation matrix, their temporal relation from the temporal matrix, and calculates the polarity relation (aligned/opposed) from the objects' internal polarities and temporal relation. These three relations then form an index into a cell within the appropriate truth table to determine consistency. Each geo-object pair is navigated to identify any inconsistent regions, which if present are output as a list of inconsistencies in the geo-model. The tool is written using the Geodes-Solutions spatial toolkit (Botella et al., 2016; Geodes-Solutions; Pellerin 2017), which facilitated spatial navigation and enabled conversion to a volumetric spatial representation where required. It was run on a moderately powerful Windows desktop, typically requiring several minutes to assess a model. Note the tool

1  is written strictly to demonstrate proof-of-concept for the framework and general approach, and is not meant for

2  widespread deployment as it is restricted to an specific, older, version of the toolkit.

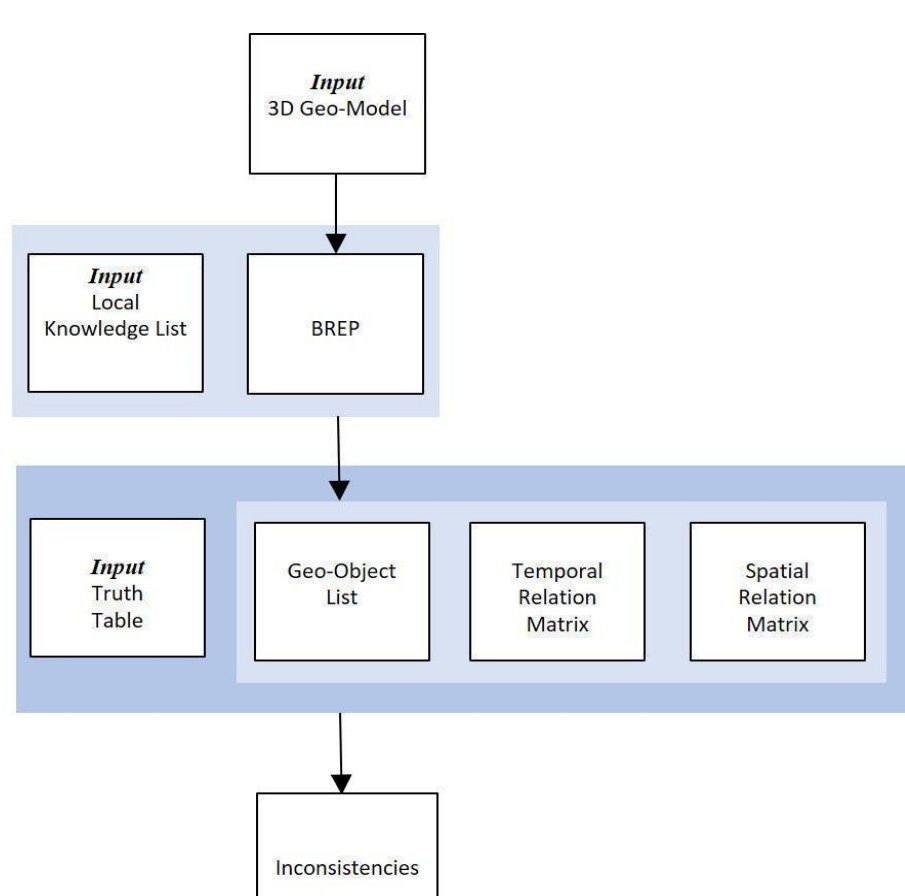

**Figure 4.** Consistency checker workflow. BREP - Boundary Representation model (Braid 1975).

**3.1 Local Knowledge List**

Local geological knowledge is specific to each study area and, alongside the 3D geo-model, is a primary input to the

consistency checker tool. It is found in a variety of sources external to the 3D geo-model, such as databases, map

legends, stratigraphic columns, journal articles and other reports. For the synthetic model shown in Figure 5a, a local

knowledge list is developed and illustrated in Figure 5b: the list contains the name of the geological object, its type

(from the nine possibilities), global internal polarity and its relative age. For simplicity in the proof-of-concept tool,

an object's global polarity is either up, down, or unknown. Also for simplicity in the tool, if a whole geological

object is an aggregate of parts, then the local knowledge applies to the whole and is assumed to be the same for

every part. For example, in Figure 5a, the local knowledge for units A and B is assumed to also hold for each of

their parts A1, A2, B1, B2; separate local knowledge for these parts, if it existed, would not be used. This enables

the spatial, temporal, and polarity to hold between the wholes, which tends to be the resolution at which the input

knowledge is available. However, nothing prevents other implementations of the framework from consistency-

checking the object parts instead of the wholes. Indeed, any simplifications in our tool should not equate to

deficiencies in the framework - they are made only to ease testing, tool-building, and presentation.

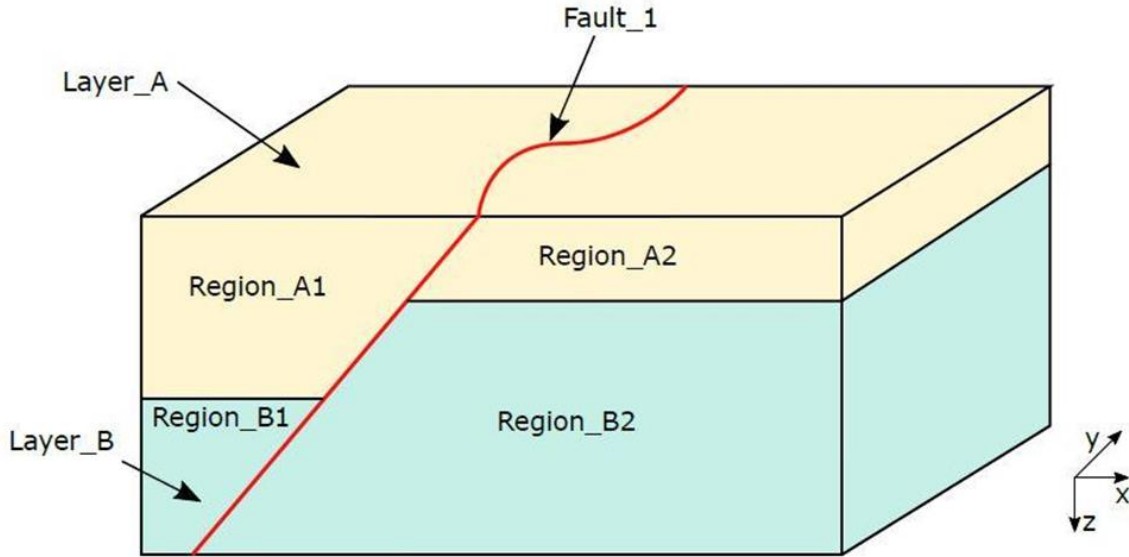

**Figure 5a.** example 3D geo-model.

| Geological entity name | Geological entity type | Geological entity polarity | Geological entity age |
|---|---|---|---|
| layer_A | depositional unit / intrusion / extrusion / metamorphic unit / fault / erosion surface / fold volume / linear fabric / planar fabric | up / down / unknown | intermediate |
| layer_B | depositional unit / intrusion / extrusion / metamorphic unit / fault / erosion surface / fold volume / linear fabric / planar fabric | up / down / unknown | oldest |
| fault_1 | depositional unit / intrusion / extrusion / metamorphic unit / fault / erosion surface / fold volume / linear fabric / planar fabric | up / down / unknown | youngest |

11                  **Figure 5b.** Local knowledge list for the model in Figure 5a.

**3.1 Temporal Relation Matrix**

Temporal relations are also obtained from external sources, including absolute or relative ages for each geological

object, as well as the kind of duration of the geological event (i.e. interval or instant). This knowledge then

determines the appropriate temporal relation between all pairs of geological objects, organized as a temporal matrix

(Figure 6). The matrix is developed manually for our case studies but could be determined automatically from

databases and other digital sources. The *incomparable* relation is chosen if there exists insufficient knowledge to

determine the temporal relation between a pair of objects.

|  | layer_A | layer_B | fault_1 | | Temporal Relations Legend |
|---|---|---|---|---|---|
| layer_A | 6 | 0 / 1 / 2 / 3 / 4 / 5 / 6 / 7 / 8 / 9 / 10 / 11 / 12 / (13) | (0) 1 / 2 / 3 / 4 / 5 / 6 / 7 / 8 / 9 / 10 / 11 / 12 / 13 | | 0 = precedes<br>1 = meets<br>2 = overlaps<br>3 = is finished by |
| layer_B | (0) 1 / 2 / 3 / 4 / 5 / 6 / 7 / 8 / 9 / 10 / 11 / 12 / 13 | 6 | (0) 1 / 2 / 3 / 4 / 5 / 6 / 7 / 8 / 9 / 10 / 11 / 12 / 13 | | 4 = contains<br>5 = starts<br>6 = equals<br>7 = incomparable to |
| fault_1 | 0 / 1 / 2 / 3 / 4 / 5 / 6 / 7 / 8 / 9 / 10 / 11 / 12 / (13) | 0 / 1 / 2 / 3 / 4 / 5 / 6 / 7 / 8 / 9 / 10 / 11 / 12 / (13) | 6 | | 8 = is started by<br>9 = is during<br>10 = finishes<br>11 = is overlapped by<br>12 = is met by<br>13 = is preceded by |

9 **Figure 6.** Temporal relation matrix for the model in Figure 5a.

**3.2 Spatial Relation Matrix**

Development of the spatial relation matrix within our tool requires transformation of a vectorized 3D geo-model into

a boundary representation (BREP; Banerjee et al., 1981). Such transformation would not be required for

implementations with alternative spatial representations or means of spatial relation determination. The BREP

ensures all geological objects are represented in their full-dimensional form, e.g. a volume initially represented by its

top and bottom surfaces is converted into a mesh of the full exterior limits of the volume, consisting of faces, edges

and vertices. A geo-model then can be traversed by following the geometric decomposition of each object and their

adjacencies. If objects are named and typed (e.g. as in the Geodes-Solutions BREP solution; Botella et al., 2016;

Geodes-Solutions; Pellerin 2017), then such traversal enables building of the spatial relation matrix. Specifically, the

consistency checker tool builds a list containing each geometric object, as well their dimensionality  (volume,

surface, or line), type (e.g. depositional unit), and name (e.g. "Layer_A"). The list is traversed in order of

dimensionality, starting with higher-dimensional objects (volumes) and progressing to lower-dimensional objects

(surfaces and lines), with spatial relations determined between pairs of objects by inspecting decompositions and

adjacencies. The results are recorded in the spatial relation matrix (Figure 7), which encapsulates the structural and

lithological topology, embedding intuitive geological relations into a computational form; other mechanisms, such

as structural and stratigraphic network graphs, may also be appropriate for representing object relations (Thiele et

al., 2016a, b). For simplicity, a cell in the spatial matrix contains a single value, and the entities being related are the

whole objects, e.g. Layer_A (Figure 5a), and not their parts, e.g. Region_A1 or Region_A2 (Figure 5a). This is

obviously problematic as distinct parts of objects might be spatially related in many ways, e.g. some might touch

and others are disjoint, so the wholes can be related in many ways too, requiring multiple values per cell for each

pair of wholes. For example it is possible Region_A1 has one relation with Region_B1 and a different relation with

Region_B2, thus A would have multiple distinct relations with B. In such cases, the most dominant relation is

selected, which suffices for our case studies. To avoid multi-valued cells, a rigorous approach would utilize object

parts for consistency-checking, rather than the wholes.

**Figure 7.** Spatial relation matrix for the model in Figure 5a.
**4 Case studies**
The consistency checker is tested in four case studies: three synthetic models in which inconsistencies are

introduced, and a real regional geo-model from ongoing project work in Western Canada (Thapa and McMechan,

2019, McMechan et al., 2021). The geo-models are built using a variety of software and underlying approaches

including: Noddy (Jessell, 1981), GOCAD/SKUA (Jayr et al, 2008; Mallet, 2004), GOCAD (Mallet, 1989) and

certain extensions, namely SURFE (Hillier, et al. 2014; de Kemp et al. 2017) and SPARSE (de Kemp et al. 2004).

**4.1 Implicit Case study**

A simple but common modelling error occurs when applying certain implicit algorithms to sparse observations of near parallel, shallow, dipping strata. Then, as depicted in Figure 1, if some older unit data are slightly topographically higher than younger unit data, algorithm bias can result in older units above younger units. To assess such a model, the consistency checker requires alignment of the three polarity vectors, two internal and one temporal: as shown in Figure 8, the CC Truth Table indicates two depositional units that spatially and temporally meet are (1) consistent, if the polarity vectors are aligned, and (2) inconsistent, if they are opposed. Therefore, this model is evaluated as inconsistent, because the temporal polarity vector is opposed to the internal polarity vectors. Note that there could be many reasons for the temporal reversal, but these are not identified by the checker, e.g. it might be algorithmic bias or missing objects, such as absent thrust faults, recumbent folds, or erosional surfaces.

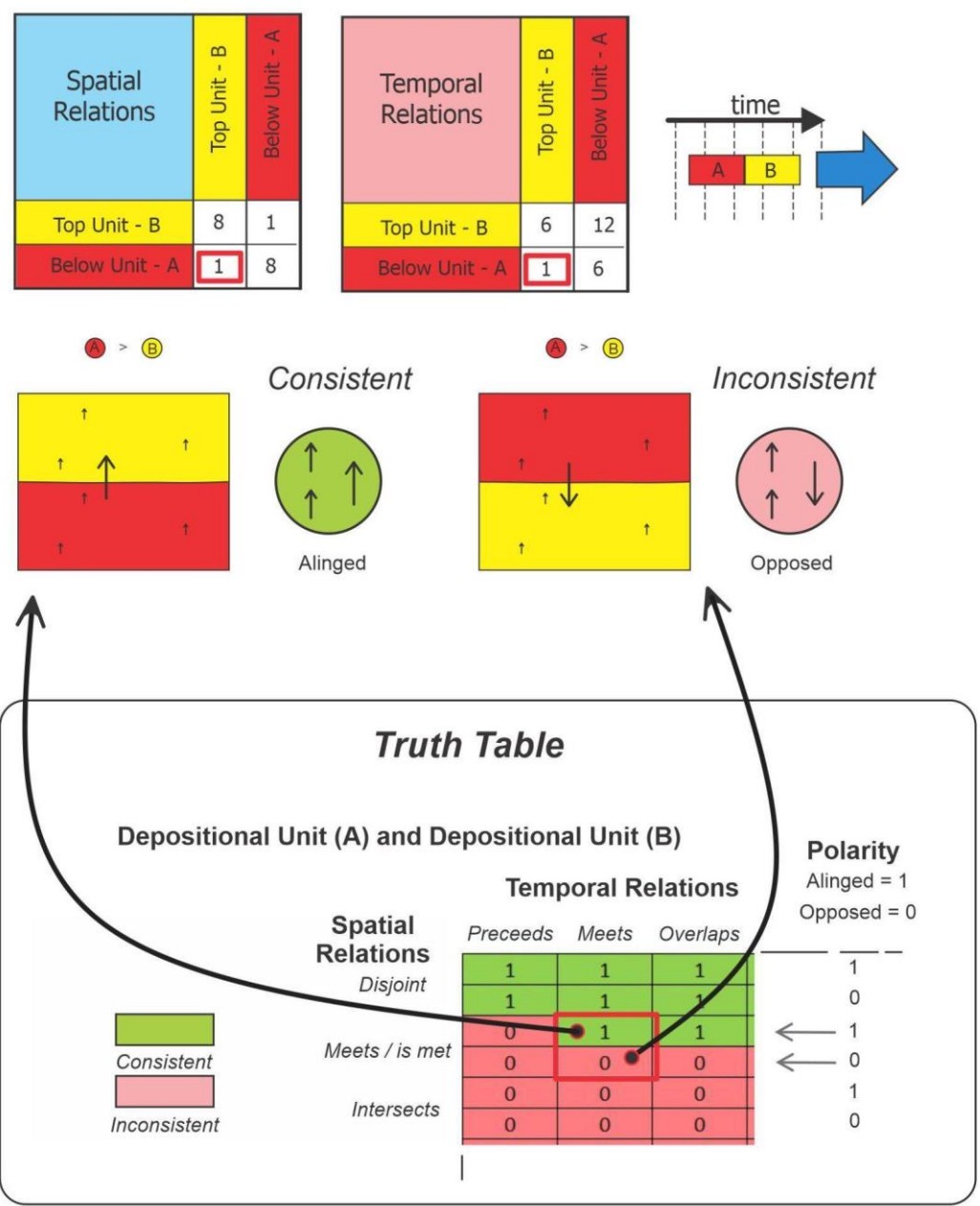

**Figure 8**. From the simple model presented in the introduction, the internal polarity vectors are aligned with each other, but opposed to the temporal vector. As the units both spatially and temporally *meet*, the CC Truth Table shows this configuration is inconsistent due to the unaligned age vector. Note that a spatial *meets* between these two units is also not allowed if there is a *precedes* temporal relation since there would be a time gap of non-deposition signaling a missing object, in fact a type of erosional surface - a disconformity.

**4.2 GOCAD/SKUA Case Study**

This synthetic model contains two depositional units, one intrusion, two faults, one fold, and one erosional surface

co-located with the top surface of the oldest unit (Figure 9). The model is created with GOCAD/SKUA (Mallet,

1989, 2004) using the Structural and Stratigraphic workflow (Jayr et al., 2008); the local knowledge list (Table 4)

and temporal matrix (Figure 10) are developed manually. The spatial matrix (Table 5) includes a variety of spatial

relations, such as touching geological units, faults cutting geological units, intrusion units protruding into other

geological units, as well as disjoint geological objects. As expected, results from the consistency checker indicate

the geo-model is geologically consistent. However, if the event timeline is manipulated to generate inconsistencies

without altering spatial relations (Figure 11), then an intersection between the second deposited layer (blue) and the

first fault (red) is detected, which is inconsistent with the altered event history (Figure 12).

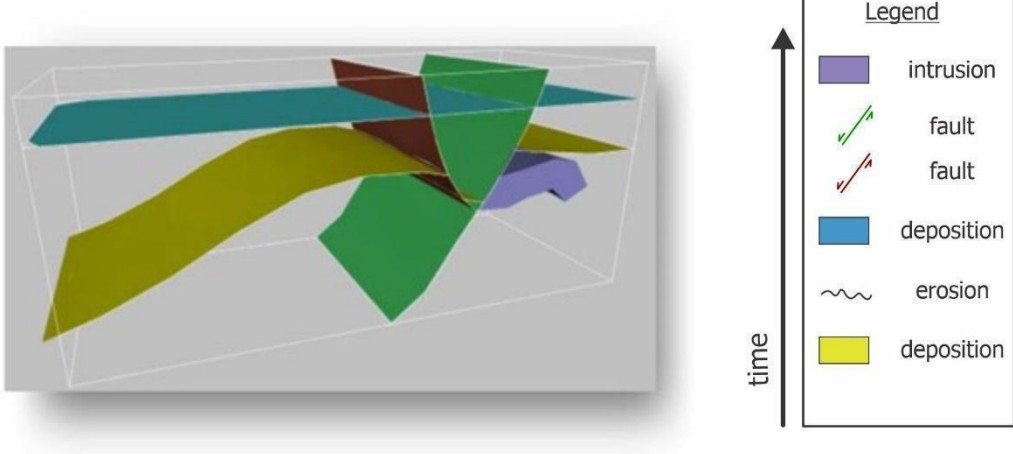

**Figure 9.** Synthetic geo-model for the GOCAD/SKUA case study: two sedimentary horizons (in yellow and blue),
one intrusion (in purple), two faults (in green and red), one fold (in the yellow horizon) and one erosion
surface (top of the yellow unit). Horizons define the top of a unit. For simplicity we ignore the folding event
that affects pre-erosion sediments.

| Name | Entity type | Age | Polarity |
|---|---|---|---|
| Intrusion_A | Intrusion Unit | Youngest | Up |
| Fault_2 | Fault Surface | Younger than F1 | None |
| Fault_1 | Fault Surface | Older than F2 Younger than HB | None |
| Horizon_B | Depositional Unit | Younger than HAE | Up |
| Horizon_A_Erosion | Erosional Surface | Younger than HA | Up |
| Horizon_A | Depositional Unit | Oldest | Up |

3    **Table 4.** Local knowledge list for the GOCAD/SKUA case study.

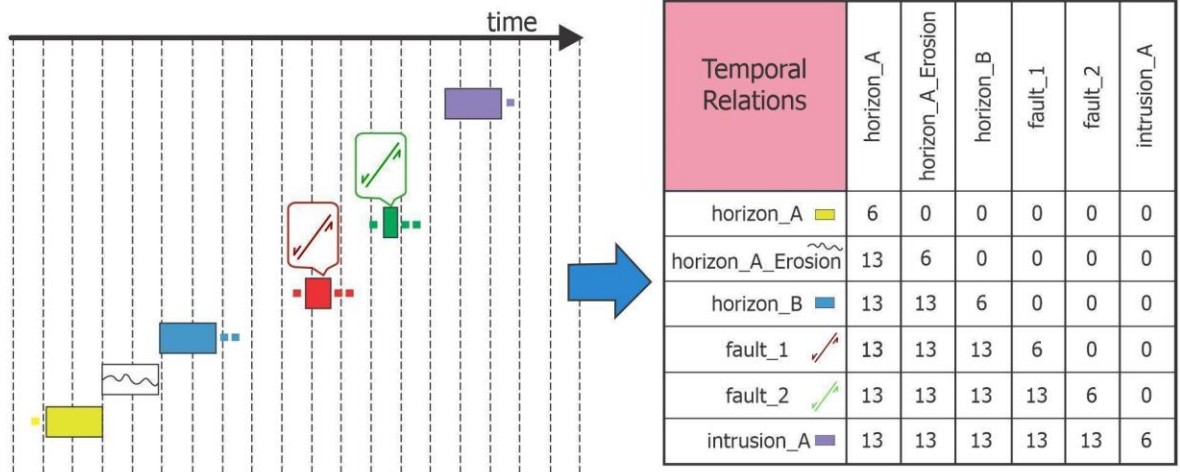

6    **Figure 10.** Event history (left) and temporal matrix (right) for the GOCAD/SKUA case study. For temporal relation
7       codes see Figure 6.

| Spatial Relations | horizon_A | horizon_A_Erosion | horizon_B | fault_1 | fault_2 | intrusion_A |
|---|---|---|---|---|---|---|
| horizon_A | 8 | 6 | 1 | 3 | 3 | 1 |
| horizon_A_Erosion | 6 | 8 | 6 | 2 | 2 | 0 |
| horizon_B | 1 | 6 | 8 | 3 | 3 | 0 |
| fault_1 | 3 | 2 | 3 | 8 | 1 | 0 |
| fault_2 | 3 | 2 | 3 | 1 | 8 | 6 |
| intrusion_A | 1 | 0 | 0 | 0 | 6 | 8 |

**Table 5.** Spatial relation matrix for the GOCAD/SKUA case study. For spatial relation codes see figure 7.

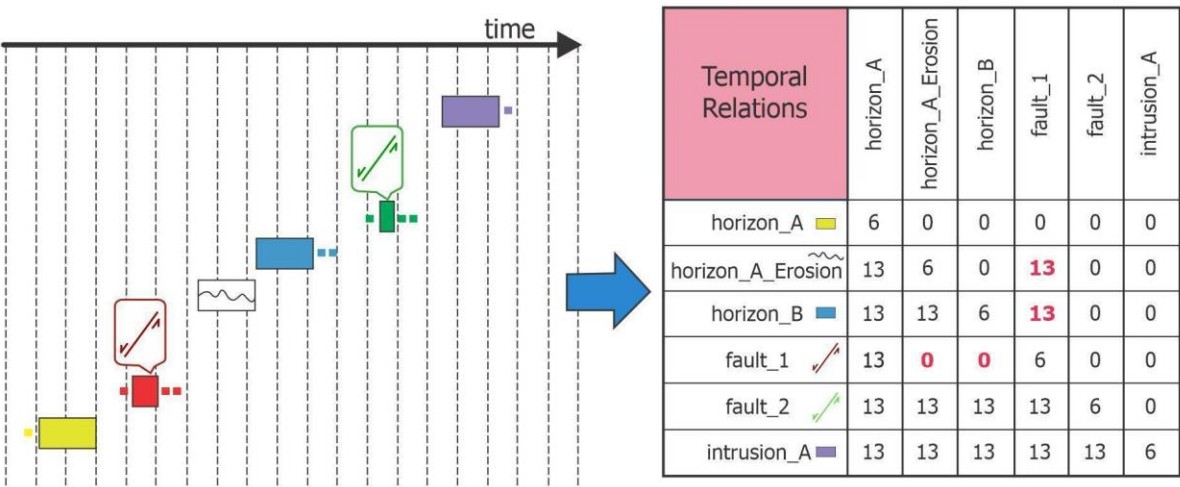

| Temporal Relations | horizon_A | horizon_A_Erosion | horizon_B | fault_1 | fault_2 | intrusion_A |
|---|---|---|---|---|---|---|
| horizon_A | 6 | 0 | 0 | 0 | 0 | 0 |
| horizon_A_Erosion | 13 | 6 | 0 | 13 | 0 | 0 |
| horizon_B | 13 | 13 | 6 | 13 | 0 | 0 |
| fault_1 | 13 | 0 | 0 | 6 | 0 | 0 |
| fault_2 | 13 | 13 | 13 | 13 | 6 | 0 |
| intrusion_A | 13 | 13 | 13 | 13 | 13 | 6 |

**Figure 11.** Modified event history (left), with red fault earlier in the event history, and temporal matrix (right) for the GOCAD/SKUA case study, with unfeasible temporal relations (in red). For temporal relations codes see Figure 6.

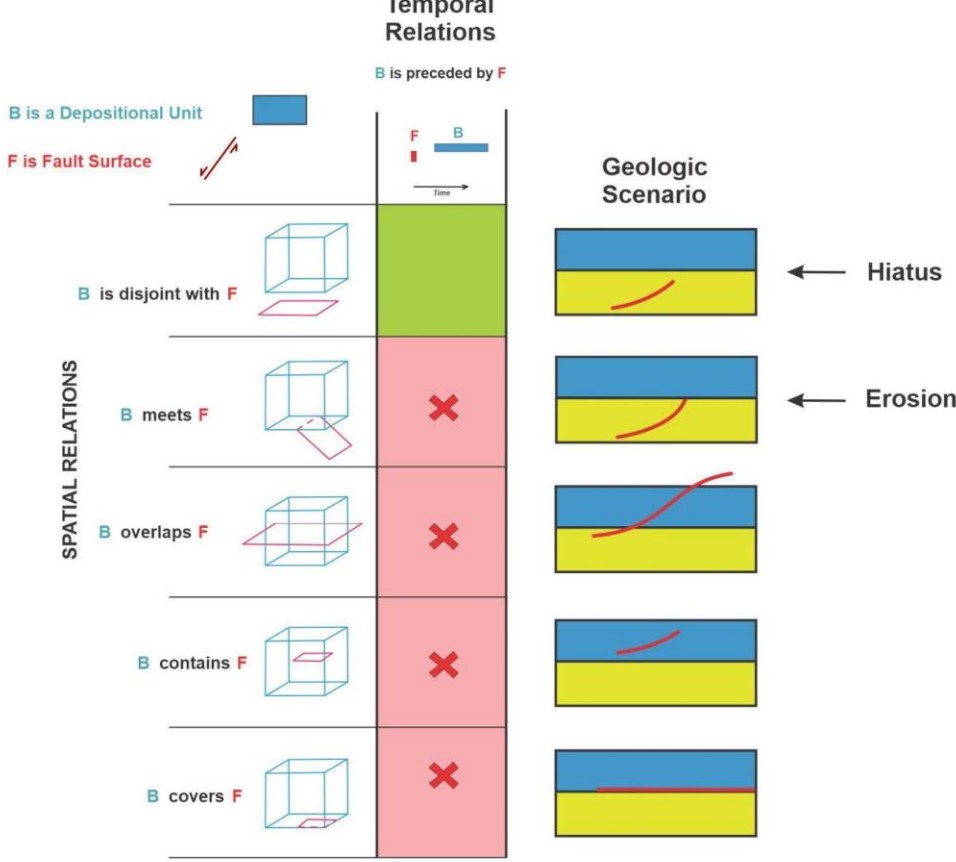

**Figure 12**. CC Truth Table fragment shows that the geological relation between F (fault_1; earliest fault in red) and B (Unit B in blue) is inconsistent with the revised geological history in which B *is preceded by* F and B spatially *overlaps* F. Other inconsistent scenarios (marked 'X') include: B *covers* F, B *contains* F, and B *meets* F. These are geologically implausible because the early fault (F) was eroded before Unit B was deposited in the altered history. The only consistent scenario, marked in green, is where F precedes B, allowing for a time gap in which the fault could be preserved in the underlying host rock. In fact, the inconsistency signals a missing object, in this case the erosional surface.

**4.3 Western Canada Case Study**

The geo-model for this real case study (Figure 13) uses data from the Rocky Mountains of the Western Canadian Cordillera, and is built using the GOCAD/SKUA, SURFE, and SPARSE toolkits (Dutranois, et al. 2010, Hillier et al. 2014, de Kemp et al. 2016). It represents a portion of an east verging fold and thrust belt that has telescoped the Paleozoic and basement meta-sediments of the early North American craton margin, with tectonic deformation having produced in-sequence and out-of-sequence thrusts (McMechan et al., 2021, Morely 1988), as well as later normal faults, with fold-fault and horizon relations that complicate original stratigraphy. The event history (Figure 14) is simplified with all the sedimentary units depositing sequentially, and incurring some facies changes across major structures, followed by several episodes of faulting with some overlapping in time. The spatial complexity of the model arises from the multitude of entities, from faults crosscutting other faults and impacting the pre-deposited layers. The resulting geometry is composed of 213 objects within the 25 units, and 6 faults, with each object delimited or separated from the rest of the unit by a fault.

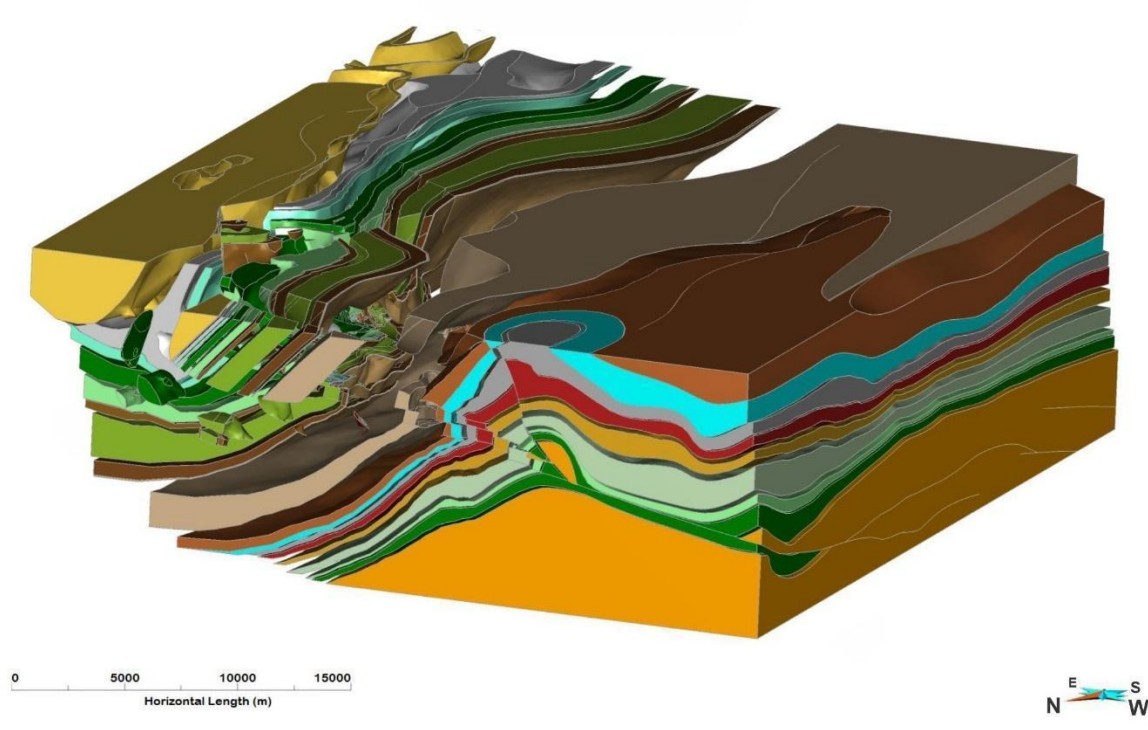

**Figure 13**. Western Canada case study volumetric geo-model: includes 213 objects with 26 geological depositional units and 6 faults. Geology from Thapa and McMechan (2019).

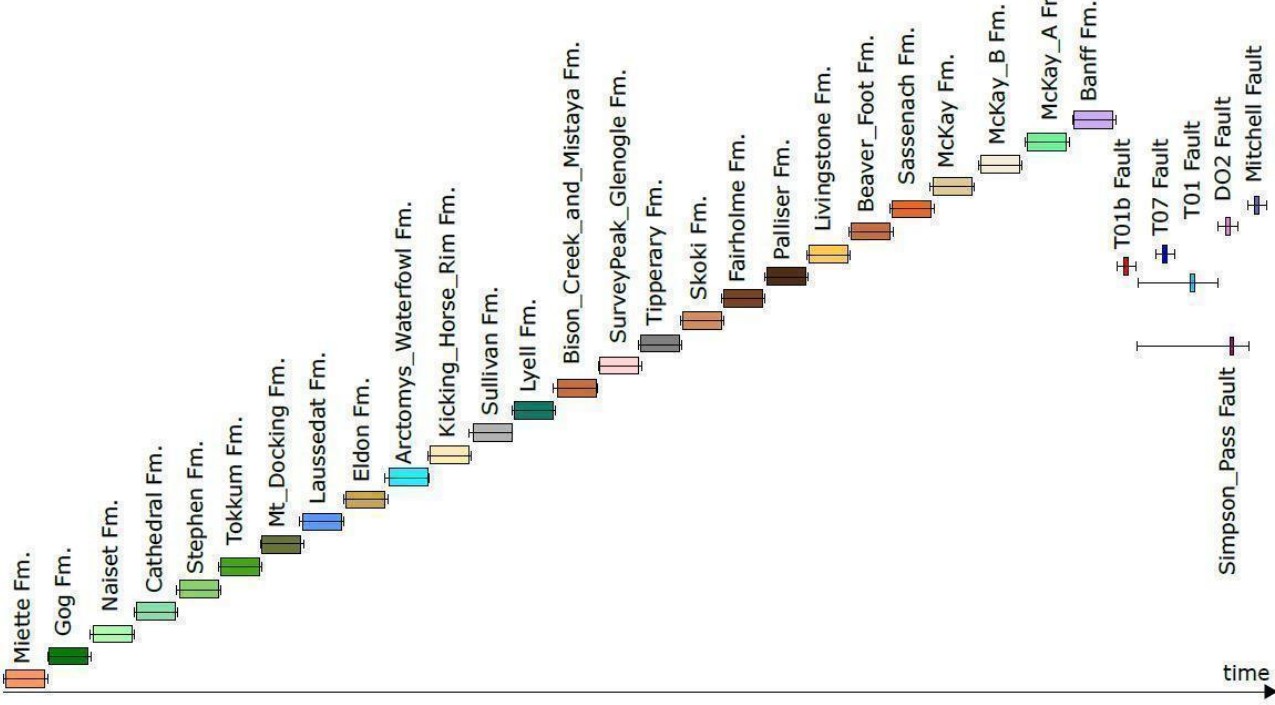

3

**Figure 14.** Event history for the Western Canada case study: horizontal boxes are relative timelines and bars are
possible ranges. Vertical axis could be used for relative spatial properties of objects such as unit thickness,
however this information was not available.

After compilation of the local knowledge list and temporal relation matrix from external sources, including maps

and reports, and development of the spatial relation matrix (Table 6), the consistency checker detects one

inconsistency. The inconsistency (Figure 15) involves the spatial containment of one sedimentary unit

(Miette:oldest) by another (Gog:younger), which is impossible given they cannot occupy the same space, being

caused by different depositional processes at different times. The consistency checker not only identifies the kind of

inconsistency through specification of the truth table cell, but it also pinpoints the location of the problem by

identifying the inconsistent volumes, after iteration through all the geo-object pairs. This error would be difficult to

detect through visual inspection alone, and if missed could have profound effect on the validity of downstream

models such as flow simulations. Subsequent analysis of the inconsistency suggests it is an artifact of the modelling

algorithm and its inaccurate interpolation of the data.

| Spatial Relations | Miette | Gog | Naiset | Cathedral | Tokkumm | Mt_Docking | Laussedat | Eldon | Arctomys_Waterfowl | Kicking_Horse_Rim | Sullivan | Lyell | Bison_Creek_and_Mistaya | SurveyPeak_Glenogle | Tipperary | Skoki | Fairholme | Palliser | Livingstone | Beaver_Foot | Sassenach | McKay | McKay_A | Banff | Kana_Topography | T01b_fault | T07_fault | T01_fault | Simpson Pass_fault | D02_fault | Mitchell_fault |
|---|---|---|---|---|---|---|---|---|---|---|---|---|---|---|---|---|---|---|---|---|---|---|---|---|---|---|---|---|---|---|---|
| Miette | 8 | 5 | 1 | 1 | 1 | 1 | 1 | 1 | 1 | 1 | 1 | 1 | 1 | 1 | 1 | 0 | 0 | 0 | 0 | 0 | 0 | 0 | 0 | 0 | 3 | 3 | 3 | 0 | 3 | 3 | 3 |
| Gog | 4 | 8 | 1 | 1 | 1 | 1 | 1 | 1 | 1 | 1 | 1 | 1 | 1 | 1 | 1 | 1 | 1 | 1 | 1 | 1 | 1 | 0 | 0 | 0 | 3 | 3 | 3 | 0 | 3 | 3 | 3 |
| Naiset | 1 | 1 | 8 | 1 | 1 | 1 | 1 | 1 | 1 | 1 | 1 | 1 | 1 | 1 | 1 | 1 | 1 | 1 | 1 | 1 | 1 | 0 | 0 | 0 | 3 | 3 | 3 | 0 | 3 | 3 | 3 |
| Cathedral | 1 | 1 | 1 | 8 | 1 | 1 | 1 | 1 | 1 | 1 | 1 | 1 | 1 | 1 | 1 | 1 | 1 | 1 | 1 | 1 | 1 | 0 | 0 | 0 | 3 | 3 | 3 | 0 | 3 | 3 | 3 |
| Tokkumm | 1 | 1 | 1 | 1 | 8 | 1 | 1 | 1 | 1 | 1 | 1 | 1 | 1 | 1 | 1 | 1 | 1 | 1 | 1 | 1 | 1 | 1 | 0 | 0 | 3 | 3 | 3 | 0 | 3 | 3 | 3 |
| Mt_Docking | 1 | 1 | 1 | 1 | 1 | 8 | 1 | 1 | 1 | 1 | 1 | 1 | 1 | 1 | 1 | 1 | 1 | 1 | 1 | 1 | 1 | 0 | 0 | 0 | 3 | 3 | 3 | 0 | 3 | 3 | 3 |
| Laussedat | 1 | 1 | 1 | 1 | 1 | 1 | 8 | 1 | 1 | 1 | 1 | 1 | 1 | 1 | 1 | 1 | 1 | 1 | 1 | 1 | 1 | 0 | 0 | 0 | 3 | 3 | 3 | 0 | 3 | 3 | 3 |
| Eldon | 1 | 1 | 1 | 1 | 1 | 1 | 1 | 8 | 1 | 1 | 1 | 1 | 1 | 1 | 1 | 1 | 1 | 1 | 1 | 1 | 1 | 0 | 0 | 0 | 3 | 3 | 3 | 0 | 3 | 3 | 3 |
| Arctomys_Waterfowl | 1 | 1 | 1 | 1 | 1 | 1 | 1 | 1 | 8 | 1 | 1 | 1 | 1 | 1 | 1 | 1 | 1 | 1 | 1 | 1 | 1 | 0 | 0 | 0 | 3 | 3 | 3 | 0 | 3 | 3 | 3 |
| Kicking_Horse_Rim | 1 | 1 | 1 | 1 | 1 | 1 | 1 | 1 | 1 | 8 | 1 | 1 | 1 | 1 | 1 | 1 | 1 | 1 | 1 | 1 | 1 | 0 | 0 | 0 | 3 | 3 | 3 | 3 | 3 | 3 | 3 |
| Sullivan | 1 | 1 | 1 | 1 | 1 | 1 | 1 | 1 | 1 | 1 | 8 | 1 | 1 | 1 | 1 | 1 | 1 | 1 | 1 | 1 | 1 | 1 | 0 | 0 | 3 | 3 | 3 | 3 | 3 | 3 | 3 |
| Lyell | 1 | 1 | 1 | 1 | 1 | 1 | 1 | 1 | 1 | 1 | 1 | 8 | 1 | 1 | 1 | 1 | 1 | 1 | 1 | 1 | 1 | 1 | 0 | 1 | 3 | 3 | 3 | 3 | 3 | 3 | 3 |
| Bison_Creek_Mistaya | 1 | 1 | 1 | 1 | 1 | 1 | 1 | 1 | 1 | 1 | 1 | 1 | 8 | 1 | 1 | 1 | 1 | 1 | 1 | 1 | 1 | 1 | 0 | 1 | 3 | 3 | 3 | 3 | 3 | 3 | 3 |
| Survey_Peak_Glenogle | 1 | 1 | 1 | 1 | 1 | 1 | 1 | 1 | 1 | 1 | 1 | 1 | 1 | 8 | 1 | 1 | 1 | 1 | 1 | 1 | 1 | 1 | 0 | 1 | 3 | 3 | 3 | 3 | 3 | 3 | 3 |
| Tipperary | 1 | 1 | 1 | 1 | 1 | 1 | 1 | 1 | 1 | 1 | 1 | 1 | 1 | 1 | 8 | 1 | 1 | 1 | 1 | 1 | 1 | 1 | 1 | 1 | 3 | 3 | 3 | 3 | 3 | 3 | 3 |
| Skoki | 0 | 1 | 1 | 1 | 1 | 1 | 1 | 1 | 1 | 1 | 1 | 1 | 1 | 1 | 1 | 8 | 1 | 1 | 1 | 1 | 1 | 1 | 1 | 1 | 3 | 3 | 3 | 3 | 3 | 3 | 3 |
| Fairholme | 0 | 1 | 1 | 1 | 1 | 1 | 1 | 1 | 1 | 1 | 1 | 1 | 1 | 1 | 1 | 1 | 8 | 1 | 1 | 1 | 1 | 1 | 1 | 1 | 3 | 3 | 3 | 3 | 3 | 3 | 3 |
| Palliser | 0 | 1 | 1 | 1 | 1 | 1 | 1 | 1 | 1 | 1 | 1 | 1 | 1 | 1 | 1 | 1 | 1 | 8 | 1 | 1 | 1 | 1 | 1 | 1 | 3 | 3 | 3 | 3 | 3 | 3 | 0 |
| Livingstone | 0 | 1 | 1 | 1 | 1 | 1 | 1 | 1 | 1 | 1 | 1 | 1 | 1 | 1 | 1 | 1 | 1 | 1 | 8 | 1 | 1 | 1 | 1 | 1 | 3 | 3 | 3 | 3 | 3 | 3 | 0 |
| Beaver_Foot | 0 | 1 | 1 | 1 | 1 | 1 | 1 | 1 | 1 | 1 | 1 | 1 | 1 | 1 | 1 | 1 | 1 | 1 | 1 | 8 | 1 | 1 | 1 | 1 | 3 | 3 | 3 | 3 | 3 | 3 | 0 |
| Sassenach | 0 | 1 | 1 | 1 | 1 | 1 | 1 | 1 | 1 | 1 | 1 | 1 | 1 | 1 | 1 | 1 | 1 | 1 | 1 | 1 | 8 | 1 | 1 | 1 | 3 | 3 | 3 | 3 | 3 | 3 | 0 |
| McKay | 0 | 0 | 0 | 0 | 1 | 0 | 0 | 0 | 0 | 0 | 0 | 1 | 1 | 1 | 1 | 1 | 1 | 1 | 1 | 1 | 1 | 8 | 1 | 1 | 3 | 3 | 3 | 3 | 3 | 3 | 0 |
| McKay_A | 0 | 0 | 0 | 0 | 0 | 0 | 0 | 0 | 0 | 0 | 0 | 0 | 0 | 0 | 0 | 1 | 1 | 1 | 1 | 1 | 1 | 1 | 8 | 1 | 3 | 0 | 3 | 3 | 0 | 3 | 0 |
| Banff | 0 | 0 | 0 | 0 | 0 | 0 | 0 | 0 | 0 | 0 | 0 | 1 | 1 | 1 | 1 | 1 | 1 | 1 | 1 | 1 | 1 | 1 | 1 | 8 | 3 | 3 | 3 | 3 | 3 | 3 | 0 |
| Kana_Topography | 3 | 3 | 3 | 3 | 3 | 3 | 3 | 3 | 3 | 3 | 3 | 3 | 3 | 3 | 3 | 3 | 3 | 3 | 3 | 3 | 3 | 3 | 3 | 3 | 8 | 2 | 2 | 2 | 2 | 2 | 2 |
| T01b_fault | 3 | 3 | 3 | 3 | 3 | 3 | 3 | 3 | 3 | 3 | 3 | 3 | 3 | 3 | 3 | 3 | 3 | 3 | 3 | 3 | 3 | 3 | 0 | 3 | 2 | 8 | 0 | 0 | 1 | 0 | 0 |
| T07_fault | 3 | 3 | 3 | 3 | 3 | 3 | 3 | 3 | 3 | 3 | 3 | 3 | 3 | 3 | 3 | 3 | 3 | 3 | 3 | 3 | 3 | 3 | 0 | 3 | 2 | 0 | 8 | 0 | 1 | 1 | 1 |
| T01_fault | 0 | 0 | 0 | 0 | 0 | 0 | 0 | 0 | 0 | 3 | 3 | 3 | 3 | 3 | 3 | 3 | 3 | 3 | 3 | 3 | 3 | 3 | 3 | 3 | 2 | 0 | 0 | 8 | 0 | 1 | 0 |
| Simpson Pass_fault | 3 | 3 | 3 | 3 | 3 | 3 | 3 | 3 | 3 | 3 | 3 | 3 | 3 | 3 | 3 | 3 | 3 | 3 | 3 | 3 | 3 | 0 | 3 | 3 | 2 | 1 | 1 | 0 | 8 | 0 | 0 |
| D02_fault | 3 | 3 | 3 | 3 | 3 | 3 | 3 | 3 | 3 | 3 | 3 | 3 | 3 | 3 | 3 | 3 | 3 | 3 | 3 | 3 | 3 | 3 | 3 | 3 | 2 | 0 | 1 | 1 | 0 | 8 | 1 |
| Mitchell_fault | 3 | 3 | 3 | 3 | 3 | 3 | 3 | 3 | 3 | 3 | 3 | 3 | 3 | 3 | 3 | 3 | 3 | 0 | 0 | 0 | 0 | 0 | 0 | 0 | 2 | 0 | 1 | 0 | 0 | 1 | 8 |

**Spatial Relations Legend:**

| | |
|---|---|
| 0 = disjoint | 6 = covers |
| 1 = meets, is met by | 7 = is covered by |
| 2 = intersects | 8 = equals |
| 3 = overlaps, is overlaped by | |
| 4 = contains | |
| 5 = is contained by | |

**Table 6.** Spatial relation matrix for the Western Canada case study; inconsistent containment relations are in red as *contains* (4) and *is contained by* (5). Depositional unit - fault relations are either *disjoint* (0) or *overlap* (3). Fault and topography relations labeled as *intersects* (2) since the model extends above topography, similarly volumetric units are in *overlap* (3) relation with the topographic surface. Most Unit - Unit relations are spatial *meets* (1) or *disjoint* (0).

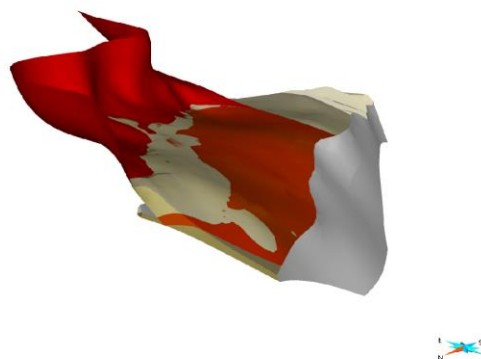

**Figure 15.** Inconsistent spatial containment between the Gog (red) and Miette (yellow) units in the Western Canada case study. The Miette is an older unit preceding deposition of Gog material, so there should not be a 'Miette *contains* Gog' or 'Gog *is contained by* Miette' spatial relation.

**4.4 Noddy Case Study**

The synthetic geo-model for this case study is generated using Noddy, which is a 3D rule-based modelling tool (Jessell, 1981) that applies an input list of geological events, or event schema (Perrin et al. 2013), to a volume of interest from which a spatial topology can be generated between objects in the volume. The event history for this case study is quite simple, including 5 major events, deposition, tilting, folding, faulting, and intrusion (Figure 16), from which the local knowledge list and temporal matrix are derived. The resulting geo-model (Figure 17) has an initial depositional sequence involving 6 depositional units (represented by their top horizons), an early tilting event followed by folding and normal faulting, and an intrusive body subsequently injected into all previous geological objects, with the fault cutting all the horizons but not the intrusion body. Navigation of the BREP representation of the geo-model yields a rich temporal matrix (Table 7) and spatial matrix (Table 8a).

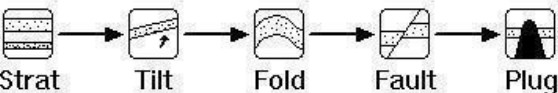

**Figure 16.** Event history for the Noddy case study: the Stratigraphic event is the deposition of 6 geological units in sequence.

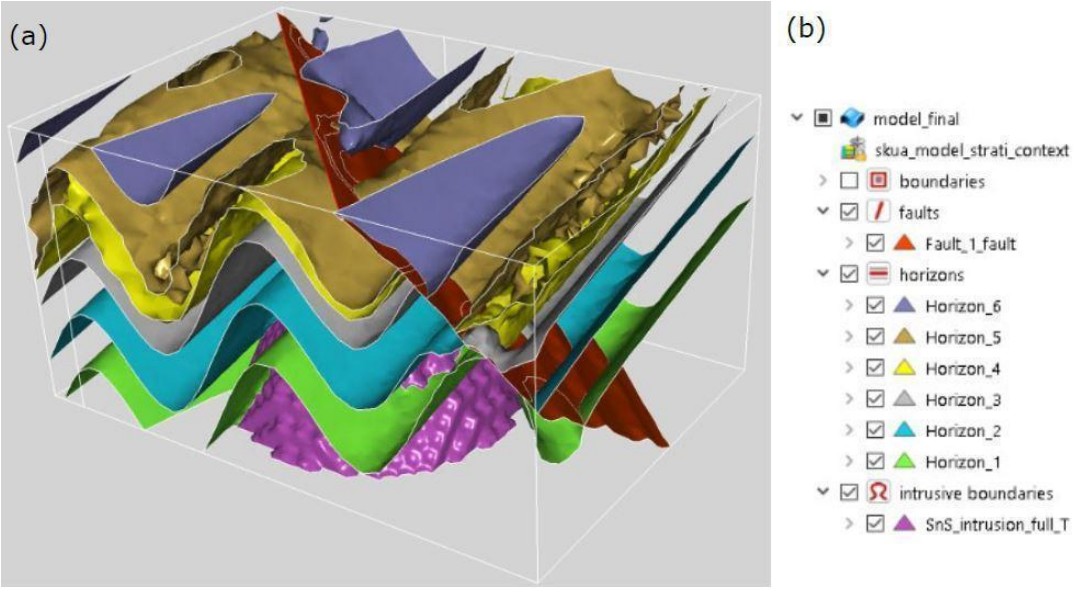

**Figure 17.** 3D geo-model for the Noddy case study.

| TIME | Horizon_1 | Horizon_2 | Horizon_3 | Horizon_4 | Horizon_5 | Horizon_6 | Above Horizon 6 | Fault | Intrusion |
|---|---|---|---|---|---|---|---|---|---|
| Horizon_1 | 6 | 0 | 0 | 0 | 0 | 0 | 0 | 0 | 0 |
| Horizon_2 | 13 | 6 | 0 | 0 | 0 | 0 | 0 | 0 | 0 |
| Horizon_3 | 13 | 13 | 6 | 0 | 0 | 0 | 0 | 0 | 0 |
| Horizon_4 | 13 | 13 | 13 | 6 | 0 | 0 | 0 | 0 | 0 |
| Horizon_5 | 13 | 13 | 13 | 13 | 6 | 0 | 0 | 0 | 0 |
| Horizon_6 | 13 | 13 | 13 | 13 | 13 | 6 | 0 | 0 | 0 |
| Above Horizon 6 | 13 | 13 | 13 | 13 | 13 | 13 | 6 | 0 | 0 |
| Fault | 13 | 13 | 13 | 13 | 13 | 13 | 13 | 6 | 0 |
| Intrusion | 13 | 13 | 13 | 13 | 13 | 13 | 13 | 13 | 6 |

**Table 7.** Temporal relations matrix for the Noddy case study. Temporal codes used here: 0 = precedes, 6 = equals, 13 = preceded by.

As a knowledge-based geo-modelling tool, Noddy will always produce a consistent model. However, export to GOCAD/SKUA via the DXF file format results in a different spatial relation matrix, one that introduces several geologically consistent but nevertheless suspect spatial relations: every unit spatially *meets* (i.e. touches) every other unit (Table 8b). Although not impossible, this is somewhat suspicious, given it contradicts the original Noddy model. As the resolution of the Noddy model is quite low, it seems likely that mesh extents might have been mis-calculated during export to GOCAD/SKUA.

a)

| SPACE | Horizon_1 | Horizon_2 | Horizon_3 | Horizon_4 | Horizon_5 | Horizon_6 | Above Horizon 6 | Fault | Intrusion |
|---|---|---|---|---|---|---|---|---|---|
| Horizon_1 | 8 | 1 | 0 | 0 | 0 | 0 | 0 | 2 | 2 |
| Horizon_2 | 1 | 8 | 1 | 0 | 0 | 0 | 0 | 2 | 2 |
| Horizon_3 | 0 | 1 | 8 | 1 | 0 | 0 | 0 | 2 | 2 |
| Horizon_4 | 0 | 0 | 1 | 8 | 1 | 0 | 0 | 2 | 2 |
| Horizon_5 | 0 | 0 | 0 | 1 | 8 | 1 | 0 | 2 | 2 |
| Horizon_6 | 0 | 0 | 0 | 0 | 1 | 8 | 1 | 2 | 2 |
| Above Horizon 6 | 0 | 0 | 0 | 0 | 0 | 1 | 8 | 2 | 2 |
| Fault | 2 | 2 | 2 | 2 | 2 | 2 | 2 | 8 | 0 |
| Intrusion | 2 | 2 | 2 | 2 | 2 | 2 | 2 | 0 | 8 |

b)

| SPACE | Horizon_1 | Horizon_2 | Horizon_3 | Horizon_4 | Horizon_5 | Horizon_6 | Above Horizon 6 | Fault | Intrusion |
|---|---|---|---|---|---|---|---|---|---|
| Horizon_1 | 8 | 1 | 1 | 1 | 1 | 1 | 1 | 2 | 2 |
| Horizon_2 | 1 | 8 | 1 | 1 | 1 | 1 | 1 | 2 | 2 |
| Horizon_3 | 1 | 1 | 8 | 1 | 1 | 1 | 1 | 2 | 2 |
| Horizon_4 | 1 | 1 | 1 | 8 | 1 | 1 | 1 | 2 | 2 |
| Horizon_5 | 1 | 1 | 1 | 1 | 8 | 1 | 1 | 2 | 2 |
| Horizon_6 | 1 | 1 | 1 | 1 | 1 | 8 | 1 | 2 | 2 |
| Above Horizon 6 | 1 | 1 | 1 | 1 | 1 | 1 | 8 | 2 | 2 |
| Fault | 2 | 2 | 2 | 2 | 2 | 2 | 2 | 8 | 0 |
| Intrusion | 2 | 2 | 2 | 2 | 2 | 2 | 2 | 0 | 8 |

**Table 8.** Spatial relations matrix for Noddy case study, including (a) original model, and (b) after export via DXF to GOCAD/SKUA. Note the replacement of many *is disjoint with* relations (**0**) in (a), with *meets / is met by* (**1**) in (b). This export operation essentially results in elimination of unit-unit *disjoint* spatial relations that will, upon re-importing into other applications, drastically distort the actual geometric relations.

**5 Discussion**

The consistency-checking framework and tool presented in this article are a first step toward the automated assessment of geological consistency in 3D geological models. The approach yields promising results in the four case studies: given minimal knowledge typically accompanying a 3D model, it detects geological inconsistencies that contravene universal geological norms captured by the truth tables. However, there is much room for improvement in determining the consistency of complex situations: the checker assesses the validity of a single geological relation in isolation, but as evident from the Noddy case study (Table 8b), a collection of relations can be inconsistent even if each relation is consistent. The consistency of such relation combinations remains a future task.

To help differentiate the various model realizations, another future consideration is the development of consistency metrics for quantitative assessment of the overall quality of a 3D geo-model. These might include a cumulative consistency score to gauge the overall effect of inconsistencies on the model, as well as perhaps targeted consistency scores for specific geo-feature relations. The latter would be particularly useful to differentiate (1) models with few inconsistencies but deep impact on internal model architecture, from (2) models with many inconsistencies but low impact on internal architecture.

Several aspects of the consistency-checking tool could be improved:

- API: development of a simple API (Application Programming Interface) to the truth tables, to enable consistency-checking from a variety of software environments, including possibly those with streamlined spatial navigation mechanisms not necessarily requiring conversion to BREP.
- Enhanced Output: from the current application or prospective API, to enhance both formatting and content, such as encoding conflicting objects using knowledge graphs or spatial standards, to facilitate visualization and understanding.

Aspects of the framework also could be improved:

- Polarity: more automated tools could be incorporated to determine polarities. The internal polarity of an object is rarely available in local knowledge, though potentially can be calculated from the modelling algorithm, e.g. as part of the scalar field gradient direction in implicit modelling, or calculation from the local normals of a

triangulated surface.  A further refinement might use local internal polarity vectors to determine polarity

relations, rather than global vectors. Supplementation from other data and methods would also be beneficial,

e.g. from various point observations, depositional top orientations and paleoflow trends, erosional surfaces,

cooling surface directions (of an intrusion or extrusion), the regional or contact metamorphic gradient for a

metamorphic unit, or directional tectonic information such as fold vergence and principal strain gradients

(Fossen, 2016; Alsop, 1999; Finkl, 1984). Fold vergence could be particularly useful: if it contradicts the

metamorphic polarity of a large orogenic terrane unit (90-180 degrees) then the situation could be inconsistent.

Generally, folds will verge away from the core or deeper axis of an orogen and these directions might be useful

in discerning juxtaposition with other objects with polarity.

● Alternate representations: it would be interesting to implement the framework on lower-dimensional

representations of geo-objects, e.g. maps and cross-sections.

● Geo-object types: consistency-checking also could be improved conceptually by expanding the list of

geological objects to include fault types (e.g. normal, reverse, strike slip) and fault domains (e.g. upper-

crust/thin-skin, deep-crust/ductile); or adding kinematic directions as another parameter in the truth tables.

These would enable, for example, comparison of macro properties such as nature of the deformation system

with the observed local kinematic conditions, e.g. thrusting or normal fault displacements.

● Parthood: as most 3D modeling algorithms and tools typically do not generate solid volumes in which one is

fully contained or covered by the other, we have set these relations as invalid for this work, knowing their

presence likely indicates a modeling problem and hence an inconsistency. However, algorithms will no doubt

mature, so future work should amend the truth tables to reflect the potential validity of such cases. This might

include further extended parameters, such as for parthood to indicate if a geological object is validly part of

another, e.g. a formation part of a group, or a fabric part of its host rock.

More generally, broadening the underlying notion of reasonableness, which thus far is roughly equated with

consistency, would yield further theoretical gains. An important assumption in the existing approach is the

correctness of input geological knowledge. As such knowledge typically reflects the understanding of domain

experts, inconsistent models often differ from the expectations of these experts (van Giffen et al., 2022; McKay and

Harris, 2016; Burch, 2003).  However, the correctness of input knowledge is a dangerous assumption, as it is more

likely that input knowledge is incomplete and has gaps, biasing expert expectations. It is necessary then to broaden

notions of geological reasonableness beyond the binary categories of consistent and inconsistent. Indeed, if we consider input knowledge might be grossly good (e.g. true) or bad (e.g. false), and models consistent or inconsistent with input knowledge, then four kinds of reasonableness emerge, as per Table 9: reasonable, unreasonable, reasonably bad, and unreasonably bad. Reasonable models, generally preferred, are consistent with good input knowledge and data constraints. Unreasonable models have geological relations inconsistent with good input knowledge. Reasonably bad models have geological relations that fit with the input knowledge, but this knowledge is wrong, or incomplete, so the model is variously questionable. Unreasonably bad models have input knowledge that may be wrong, and the model is also inconsistent, because of algorithm bias, scale/resolution, constraint data configuration or other processing errors. Inconsistent models thus signal a need to adjust the algorithm or investigate the input data and knowledge. Note, however, all models might be useful (Gleeson et al, 2021), as any geo-model from bad knowledge might be preferred to no models, or models with no input knowledge; and an inconsistent model from good knowledge, that is unreasonable, might be preferable to the alternatives, especially in parts where it is actually consistent.

## Model Consistency

|  | Inconsistent | Consistent |
|---|---|---|
| **Good Knowledge** | Unreasonable | Reasonable |
| **Bad Knowledge** | Unreasonably Bad | Reasonably Bad |

**Table 9**. Types of 3D geo-model consistency.

It is also noteworthy, and sobering, that an ideal model - i.e. one close to reality and matching input data - could arise from any of the four categories, simply because the combination of input knowledge, data, and computational processes just happens to produce the best result. Consistency-checking thus provides only some insight as to whether an ideal model is achieved, as one would hope an ideal model should be consistent more often than not. For example this should be the case when comparing a suite of models and their flow characteristics, with 'reasonable'

models matching the real world historical production curves (Melnikova et al., 2012). Mounting evidence suggests even a minimum of geological knowledge and improved consistency with this knowledge can improve the utility of models (Giraud et al., 2020, Bond et al. 2015). Enhancing our ability to embed this knowledge into 3D workflows will be an ongoing and important task to increase potential for developing more reasonable geological models (Maxelon et al. 2009).

Finally, application of the framework to case studies at various scales, using different tools and algorithms, would provide further insight into its utility for: exploring different levels of geological and model complexity (Pellerin et al., 2015); comparing high-resolution to generalized regional models; testing more speculative models; for correlation of jurisdictional bordered models (e.g. comparing number and variety of entities, and consistency with each other); and finally for assessing the range of possible 3D geological models created from probabilistic and future generative AI methods.

## 6 Conclusions

Due to the increasing complexity of current geo-modelling algorithms, leading to a plethora of models of variable quality, there is a clear need for a quick and easy-to-use approach to check the geological consistency of a model. The consistency checker framework and proof-of-concept tool developed in this paper successfully verify geo-models in four case studies, confirming consistencies and finding inconsistencies. Inputs include knowledge typically available with any geological model, namely, the spatial-temporal-polarity relations between pairs of geological objects. A specific combination of these inputs serves as an index into a CC Truth Table to document a possible geological situation that is either consistent or inconsistent with established geological principles. Altogether, this work represents a first step toward the real-time consistency-checking of geo-models; therefore, it is also potentially a first step toward interim consistency-checking during model-building, to help increase knowledge constraints in geo-modelling algorithms.

**Appendix A**

**Algorithm 1: Consistency-checking**

**Require:** *Mspatial* spatial relationship matrix, *Mtemporal* temporal relationship matrix, *LNaturePolarity* nature and
    polarity matrix of entities, *Truthtable* 'Truth Tables' for each pair of geological entities, *LGeologicalEntities* list
    of all geological entities detected in the 3D model.
**Initialize (empty):** *Linconsistencies* list of inconsistencies detected inside the given 3D geological model
    **for** each *GeolEntity* in *LGeologicalEntities* **do**
        **for** each *GeolEntity* in the remaining rows in *LGeologicalEntities* **do**
            extract the name, nature and polarity in each geological entity from *LNaturePolarity*
            given both geological entities, find the corresponding truth table *Truthtable*
            deduce the polarity relation from both geological entities: aligned, opposed, unknown
            extract the spatial relationship for the pair of geological entities from *Mspatial*
            transform the spatial relationship into a row in the truth table
            extract the temporal relationship for the pair of geological entities from *Mtemporal*
            transform the temporal relationship into a column in the truth table
            **if** the statement found in the corresponding truth table is 'inconsistent' **then**
                **for** each part in each *GeolEntity* **do**
                    **for** each part in each *GeolEntity* **do**
                        extract the name, nature and polarity of each geological entity inside *LNaturePolarity*
                        given both geological entities, find the corresponding truth table *Truthtable*
                        deduce the polarity relation from both geological entities: aligned, opposed, unknown
                        extract the spatial relationship for the pair of geological entities from *Mspatial*
                        transform the spatial relationship into a row in the truth table
                        extract the temporal relationship for the pair of geological entities from *Mtemporal*
                        transform the temporal relationship into a column in the truth table
                        **if** the statement found in the corresponding truth table is 'inconsistent' **then**
                          **for** each part in each part of *GeolEntity* **do**
                            **for** each part in each part of *GeolEntity* **do**
                              etc...
                          **end for**
                        **end for**
                      **end if**
                  **end for**
                **end for**
            **end if**
            add the statement found in the corresponding truth table to *Linconsistencies*
        **end for**
    **end for**
**return** *Linconsistencies*

1    **Appendix B** Detailed Deposition- Erosion consistency checking examples.

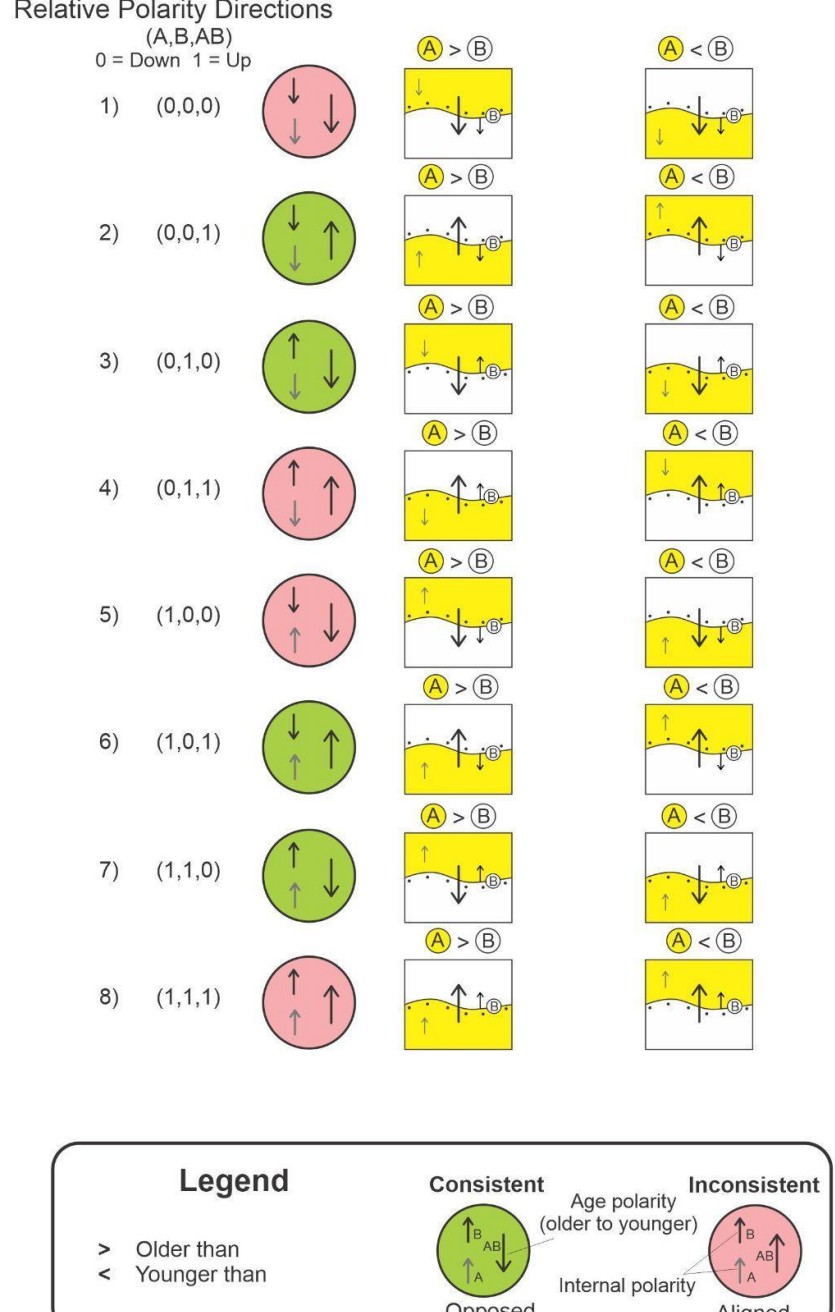

Polarity configuration examples for geological relations between **Depositional Unit (A) - Erosional Surface (B)**.
Spatial *covers* and temporal *precedes* are used.  Note that Case 1) and 8) are equivalent as well as Case 4) and 5),
and these are inconsistent. Case 2) and 7) are equivalent as well as Case 3) and 6), all are consistent. Case 7 (A < B)
is perhaps an end member consistent case, for example a Karst cavern ceiling being sealed (covered) with sediment
from the bottom to seal the roof.

## Code and Data Availability

Consistency-inconsistency matrices, called CC Truth Tables, used for determining validity of geological spatial-
temporal relations; https://doi.org/10.5281/zenodo.13948382  last access: 17 October 2024. (de Kemp, 2024).

## Video Supplement

11 There are currently no video files (mp4) related to this article.

## Author contributions

13 Conceptualization by MP, EdK, BB and MH; MP developed the system; MP, EdK, BB and MH all contributed to
14 the case studies development and the writing of the paper.

## Competing interests

The author declares that there is no conflict of interest.

**Special Issue Statement:** This contribution is part of the Loop stochastic geological modelling platform –
development and applications, edited by Laurent Ailleres
(https://gmd.copernicus.org/articles/special_issue1142.html, last access: 6 April 2024).

## Acknowledgements

Generous support for this research was provided from the Canada3D project (C3D), through the Open Geoscience
Initiative of Natural Resources Canada (https://canada3d.geosciences.ca/, last access: 6 April 2024). Support from
the LOOP project (https://loop3d.github.io/, last access: 6 April 2024), Australian Research Council (ARC);
(Enabling Stochastic 3D Geological Modelling, LP170100985), in collaboration with the OneGeology initiative is
gratefully acknowledged. Thanks to the many collaborators from the LOOP team including Mark Jessell (UWA) for
providing the Noddy model used for the first application of the consistency-checking tool. Many thanks to
RING (https://www.ring-team.org/, last access: 6 April 2024) for academic support for use of GOCAD/SKUA
software, and  Geoid-Solutions libraries (https://geode-solutions.com/, last access: 6 April 2024) for model format
conversions from GOCAD/SKUA (Model3D) to VTK.  Use of Leapfrog Geo graciously provided by Seequent.
Finally, we graciously thank all our reviewers; Rob Harrap, Sam Thiele and Michel Perrin, for taking the time to
improve our work. This paper is NRCAN contribution number 20230104.

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
