# Peer review of "Consistency-Checking 3D Geological Models"

_EGUsphere, 2024_

## Referee Comment (RC1)

Review. Parquer, de Kemp, Brodaric, Hillier
Consistency Checking 3D Geological Models

p2 l3
        ok.

16   I'd argue the same could be said for human-led compilation efforts
     where   a) # of relationships " large
             b) # of contacts that imply/require a geological order " large
        Consistency is still consistency if you just have 1 human
        making a map.

        This leads to line 2 on p4: with a truly complex situation
        (# rel'ns, # poly) is manual inspection likely?

p4 l4   implicit knowledge can vary... yes, I agree, but to introduce that
        here... are you sure no step in the model population process
        didn't also? And perhaps component data did too?

        ↳ line 7... and designed to limit/remove implicit?

l16   (not true but also not published → just an aside → using)
      ArcView/Avenue the Harrap 2001 method was
      applied to checking 600 maps (with introduced
      errors)... and of course never published :.

l17   range of knowledge. Unclear
      range of knowledge (for example,... )   perhaps.

l25   I would argue now (having read Burns after 2001) that
      his vision of using it as a tool to formalize knowledge
      in large areas → consistency while building → is
      very important. He just never thought to call it a
      legend language, but note his "chain diagrams" are
      well on their way to my ideas on graph rewriting.

PS.    An aside, but fyi, the procedural generation community
(video game world.) is beginning to look at geological history / 3d
volumes / etc. vis a vis applying terrain erosion algorithm.
(Guillaume Cordonnier...) and thogh their goals are different,
in the long run this is another form of consistency work.

eg Gailleton et al 2024 Chonk 1.0 : landscape evolution
framework: cellular automate meets graph theory
Geosci Mod Dev 17, 71-90.

In your intro you almost nail down what a consistency-checking framework
could be. I'd argue (and perhaps you disagree... which is fine) it has at
least 3 uses          ) as a testing framework (the obvious case)

2) as a generative framework: can you generate alternative
geological histories that are consistent with the data, or
conversely, given a history in the singular sense can
you generate permissable (testable) map relations
(in Bums these would be testable binary relations,
in Harrap these would be adjacency relations
that are consistently permissable in a GIS).

3) As an artifact that can be shared as an explicit
extension and formalization of geological legends
(I emphasize this but I'd argue Bums is getting at
the same idea implicitly with his "chains").

Are you covering all of these (if you care to?)

Figure 1:  I don't object to the figure. I'd argue that you do need to
include in the body a brief discussion of how these
errors "happen"        - are they in the black box of method?
                        - do they in part arise because of the
                          specific placement of data  ie
                          moving a point might testably (non black box)
                          have consequences
                        → a typology of errors is obviously too much, and likely
                          impossible, but a sentence (even if just to say "too complex"?

RMH   Paquer paper p3.

p5 21-25   This is weak. A lot has been written on this. I'd either contract
           this to a sentence or do it justice by expanding it.

p6   4,5      geological "area". domain? area of common history?

     6        interpretational → interpretive?
              process history. A geological process history includes...

     9        , and rock type classification

     18       An aside, but I'd argue that at least some mapping is intended
              to identify contradictions or gaps in extant geological theory.
              In this case a contradiction is desirable in the short term.

     22       as well as foundational...

     22       Truth Tables.-   by jump here.    "these relations"?
     ↓             ↓                            what is an object?      )  needs expansion
     26       should not be capitalized.                               )  and clarification

p7   14-17    give an example of such a global/local disparity?
              Rafter polarity in (16).

     24       It points roughly   not exactly   (can be 30° off,...)

     27.28    if you are in the subsurface beneath a vent
              is it extrusive? (nit. pick, but...)
              For this whole section I'd define vectors as roughly/approximately...
              pointing...

p8   10-17.   This is tricky. Knowledge of most of the earlier
              vectors was grounded in "locally observable field relations"
              In some cases metamorphic gradients may be quite abstract
              (they are not locally testable).
              Is metamorphism an event? Or a mappable unit?
              For example, is a local marble from regional metamorphism
              part of, say, a "volume that is at lower amphibolite grade"
              or is it a "marble". What is a metamorphic "unit"??

18   picky, but some structural geologists would map a fault volume,
      eg a shear zone.   or even brittle anastomozy networks
      Just as you exclude kinematics, you may want to exclude fault
      "volumes".

      erosion surface (hence my reference to Cordonnier's early-stage
                        work from that perspective.

      fold volumes: again, abstract volumes. I get this is a
                     placeholder but this is going to be very messy.

9    5, 9.   Fabric. No, really elements? A fold hinge ≠ fabric to me.

15-16.   Just an aside here, but if you are going this direction then
         in fairness you should cite Bems 1969, 1975 in your intro as
         he very directly addresses the fabric/time relation in his
         consistency work

20-21   Again a big leap internal to a paragraph.
        Perhaps 1-2 s talking about topological relations like
              meets etc then use your examples as is?

25.26   reword. Also possibly expand. The current phrase implies
        this is the role. Is it? Or is it a role.

10.    Perhaps in caption refer to what the volumes are in (d)
       Speculative/arm waving: in e, a fault has ↑ because it
       must be younger than bounding units. I see, though, why you gave this.

-13-

p15 l 18   "Truth Tables" Argh ∷

16   f3.   h. I dislike the arrow on fault. Perhaps ↓↓↓/↑↑↑ and ↑↑↑/↓↓↓ ??
              It is not wrong but the singular arrow
              took time to mentally attribute...

BMH   Paper 5

17   17→28: Not convinced I'd lump these all as "stratigraphic"
         paleo: ... unless otherwise indicated?
                  (many fossils are general.).        not picky.

31.   Cute math but without context. – either expand or remove.

p18      Truth Table. Again. Not going to rant again ☺.

13       here you pin down met. units as non. abstract (unlike fold
         volume).

p 19   28-29. Ok, here, you do address my comment on p5 re generative
         frameworks. I agree that as volume incompleteness of the
         space of models (well, combinatorics. – yikes).

p20    20     Geodes – Solutions : is that a URL? Citation standard check

p 23   F6      Would be clearer if you just listed temporal relation not all #'s.

       8 –10.    I guess. There is another way to look at this using a 3d extension
                of topology as seen in GIS. To illustrate in 2d.

       so perhaps expand
       8 10 a bit: what "
       actually happens)?
       What is big O consequence
       of the species?

       Really we just need a
       graph that traverses the grid:
       the dual graph of the (in this case 2d)
       topology, with labelled edges

       Not sure if you can
       collapse binary relations,
       that mesh is needed.
       Of course, if you don't have labelled
       boundaries... –

p24
       F7    same complaint as F6.

29    F12    Truth table fragment

           'see, it looks better without Table ☺.

30    F13    Approximate scale of volume.

31    13-14    Is this area "inside" the volume? If so the chance of
           it being "noticed" or even "found" by inspection if an
           error was known but not situated is very small.

59    1-2    My gut feeling (...) is that this may also address
           situations of under-constraint: while very many
           models are possible, geological knowledge
           also suggests which are plausible. Dangerous
           ground to tread, though. Some people are
           working in that general area (C. Bond in
           particular). Epistemology of field geology...

General concluding comments

1) I'd like to see a very short (2-3 s?) appendix that
provides more info on development, probably part of
A1, in the 'Code and Data Available' area. The toolchain
discussion in the paper is understandably terse.

2) While my notes on your intro are rather loose
I do think tightening up the     generate      idea is
                          /\          worth doing.
                    test — communicate

I realize 'communicating legends" etc is not your goal but
in many cases of complex models like your last case study
pebkac and... well... that's a communication issue.
I get your truth table approach, but is not elegant
in tens of immediately graspy relations (feel free to
disagree, I spent 6 hours on this not weeks, but tbh
are your uses more in the "few hours" or the "intense study"
camp?

3) Undoubtedly a "preprint" issue but a lot of figures (1,2) could
benefit from larger block models using the available space.

                                                        //RMH June 2024.

---

## Referee Comment (RC2)

[revised manuscript text omitted]

*more important are the time & visualisation challenges*

[Figure]

Important why? please expand.

2005), such as adjacency, inclusion or intersection. An important aspect is the dimensionality of the spatial objects, which might be 0D (a point), 1D (a line), 2D (a surface), or 3D (a volume). Spatial relations between such objects have been widely examined, with distinct relationships identified between 2D regions (Egenhofer and Franzosa,

1991) as well as 0D, 1D, 2D, and 3D regions (Zlatanova et al., 2004). They also have been applied to certain types of geological objects (Schetselaar and de Kemp, 2006), providing a basis for the spatial component of geological knowledge, and underpin efforts in knowledge-driven 3D geological model construction (Zhan et al., 2022; 2019).

However, they are not yet applied to the evaluation of geological models, especially in combination with temporal relations, despite being applied to the evaluation of models in other domains (e.g. Van Oosterom, 1997; Gong and

Mu, 2000; Arora et al., 2021; Nikoohemat et al., 2021; Bezhanishvili et al. 2022).

Arguably not true: many software check if e:s a generated mesh is "watertight".

[revised manuscript text omitted]

*one or more instantaneous displacements*

and associated with  (Figure 2e). The surface lacks

*what about hangingwall vs footwall concepts and their link to age & offset??*

internal polarity, as it is never constituted by any material. We are not, at this stage in our approach, considering polarities from kinematic properties distributed on faults such as slip displacement directions.

● Erosion surface: a spatial surface where a rock volume has completely eroded via a mechanical process. It is associated with a time interval or instant indicating the end of the erosion process. Its gross internal polarity points towards the eroded unit, in the direction of material destruction (Figure 2f); typically, this direction will be opposite to that of an overlying sedimentary unit.

● Fold volume: a rock volume affected by ductile tectonic deformation processes with an associated time interval.  The fold volume can be composed of depositional, intrusive, extrusive, or metamorphic material, since these units can all become folded, as can an earlier fault or erosional surface. The gross internal polarity points in the direction of structural vergence or tectonic transport direction (Figure 2g). For our

[Figure]

[Figure]

*Consider renaming to "deformation volume" or "strain volume"?? or "tectonic volume"*

purposes, nappes, duplex structures and orogenic scale tectonic units could be classified as fold volumes.

They may contain multiple folding events and have fold-fold relations. We do not treat these in this study, but consider they could be dealt with using a similar consistency approach. For example, an older fold set cannot re-fold a younger generation of folds.

[revised manuscript text omitted]

*Handwritten annotations:*

*Does A contain B in both cases? Surely these situations are topologically different.*

*⟹ Cannot both be "contains".*

*Alteration or Metamorphism:* [A & B occupy the same space]

*Intrusion: A and B cannot occupy the same space.*

*Give a geological twist?*

*"overlies"*

*"Intrudes"*

*"Abuts"/"Truncates"*

*"Overprints"*

*Consider renaming to avoid confusion [given the geological meaning of those terms.]*

[Figure]

**Table 1.** The 9 spatial relations between two geological objects of 1/2/3 dimensions. Blank gray cells denote impossible spatial relations after Egenhofer (1989); Egenhofer et al. (1993); Egenhofer and Franzosa (1991); and Zlatanova et al. (2004).

[Figure]

**2.3 Temporal Relations**

Temporal relations are required to establish a temporal ordering between geological objects (Perrin et al., 2011).

Though the temporal position of a geological object is not always known (Michalak, 2005), the temporal ordering between objects can be derived from the timeline of associated generative events (Galton, 2009; Claramunt and

Jiang, 2001). As with spatial relations, dimensionality plays a role: temporal relations can be categorized according to the nature of the time duration (of the event) with 3 potential combinations: period/period, period/instant, or instant/instant. Building on Allen's definitions (Allen, 1983), this leads to 14 distinct temporal relations, including converses (e.g. A *precedes* B and B *is preceded by* A), as shown in Table 2, for the nine geological object types; moreover, for any pair of objects only one temporal relation can hold. Of note is the *is incomparable to* relation, which indicates the temporal ordering is unknown due to unavailable temporal knowledge about one or both objects.

Tuple Equation 2 illustrates the three-part tuple for expressing these relations. The symmetric relations are *equals*

and *is incomparable to*, with the remainder being asymmetric.

$$
Entity_A \left\{ \begin{array}{l} precedes \\ meets \\ overlaps \\ isfinishedby \\ contains \\ starts \\ equals \\ isincomparableto \\ isstartedby \\ isduring \\ finishes \\ isoverlappedby \\ ismetby \\ isprecededby \end{array} \right\} Entity_B \qquad (2)
$$

[Figure]

| temporal relations | period / period | period / instant | instant / instant |
|---|---|---|---|
| A *precedes* **B**
 **B** is preceded by A | | | |
| A *meets* **B**
 **B** is met by A | | | |
| A *overlaps* **B**
 **B** is overlapped by A | | | |
| A *starts* **B**
 **B** is started by A | | | |
| A *finishes* **B**
 **B** is finished by A | | | |
| A *during* **B**
 **B** contains A | | | |
| A *equals* **B** | | | |
| A *is incomparable to* **B** | | | |

*"follows" and "followed by" may be better, to avoid confusion with spatial terms?*

**Table 2.** The 14 temporal relations between two geological objects, after Allen (1983). The temporal timeline advances from left to right in each cell. Blank gray cells denote impossible temporal relations, and blank white cells denote unknown temporal relations.

**2.4 Polarity Relations**

A polarity relation can be determined from up to three independent component polarities (discussed earlier in Section 2.1); the two internal polarities, dependent on the type of geo-object and processes governing its formation, and the age polarity. The internal and age polarity vectors, can be compared to determine if they are 'aligned' or 'opposed', such that:

- Aligned polarity relation: the vectors are roughly parallel, such that each vector is within 90° of every other vector.

- Opposed polarity relation: a vector is oriented in an opposite direction to the others, such that one vector is at least greater than 90° from one of the others.

*Clarify that this is easily calculated as the sign of the dot product*

If an object does not have internal polarity, such as a fault, or the internal polarity is missing due to lack of knowledge, then the polarity relation defaults to aligned. Importantly, polarity alignment or opposition does not

necessarily determine (in)consistency alone, as such determination requires consideration of the spatial and temporal relations. For example, opposed internal polarity can indicate either inconsistency or consistency: e.g. touching

*use appropriate term from Table 1 (meeting?)*

depositional units with opposed internal or temporal polarities are inconsistent (Figure 3b), because such units must create material in the same spatial and temporal direction; but a touching depositional unit and erosional surface with opposed internal polarity are consistent, because the surface must erode material towards the older unit (Figure

3e). The internal polarity relation also might not play a determining role in assessing (in)consistency, as the spatial and temporal relations may individually or together be determining factors: e.g. the (in)consistency of an intrusion into a host depositional unit can be determined without regard to internal polarities (Figure 3c-d), as the intrusion must be younger and touching the unit, otherwise some interceding object such as a fault or erosional surface is missing from the model; similarly for a depositional unit and a fault surface (Figure 3g-h), as the unit must pre-exist the fault. Note *that* we do not consider growth faults to be strictly synchronous within a full unit, since at least some of the material needs to pre- *date* the faulting. On the other hand, there may not be any age polarity information available other than what can be derived from the internal process vector directions. In the case of intrusions this can be observed from cooling margins, inclusions along contacts and other textures indicating internal growth and assimilation direction. Also note cases needing all three polarity components might be unaligned if the age polarity is unaligned with either of the internal polarities (see Figure 1,c-f). A general framework for (in)consistency therefore must consider the spatial, temporal, and polarity relations.  This is accomplished by using these relations as an index into 'Truth Tables' representing geological norms and specifying the (in)consistency of the situation (see

Section 2.5).

\* therefore units deposited immediately above a depositional unit do not touch/meet the units below it (e.g. overturned basement strata?)

C (young)    +  +     A (old)
         + B +

... Does B "meet" D?
D   ↑    unconformity
A / B / C

[revised manuscript text omitted]

---

## Author Comment (AC4)

**Final Author responses to GMD reviewers**

Paper:
Consistency-Checking 3D Geological Models, EGUsphere [preprint], https://doi.org/10.5194/egusphere-2024-1326, 2024.

We first provide some general comments addressing issues variously identified across all 3 reviewers. Detailed responses to each reviewer follow these general comments, with our responses marked in red font. We intend to clarify all these aspects in the revision. We thank all reviewers for their insightful comments, and agree with many, which will lead to modifications in the text. We disagree with Reviewer 3 that certain limitations are "severe", and provide a detailed rebuttal for each. Essentially, we hold that our novel framework will function appropriately for most common geological situations and current modelling paradigms, and the supposed  limitations are either (1) valid simplifications introduced to ease testing, or (2) additional features that can be progressively added in future work. We also hold that partial testing of the framework is reasonable, given the size of the problem, leaving to future work the refinement and testing of the remainder of the framework.

1.      **Paper scope:**  the paper proposes a new framework for consistency-checking 3D geological models (geomodels), and tests a portion of the framework using a proof-of-concept tool and several case studies. The framework consists of a hyperspace of all possible and impossible geological relations, between 9 geological object types, against which relations in a model (or map, x-section, etc.) can be verified for consistency. In a sense, this hyperspace strives to encapsulate the rules and principles of geology, using spatial, temporal, and polarity relations as axes for this space, as well as an axis for the related object types. The framework also could be seen as a relation-based meta-structure for geology in addition to a hyperspace. In any event, we acknowledge the framework requires expansion (e.g. at least for parthood and for more object types), and that we successfully test only a portion of the hyperspace in several case studies (4 from 9 objects [depositional and intrusion units, and faults and erosional surfaces], and 6 from 45 subspaces). But we do think this portion includes very common situations. Therefore, we feel the paper sufficiently demonstrates the viability of the framework, with the remainder left to future work. The problem is quite large (i.e. modeling all possible geological relations / situations as per the rules of geology), so we understand its full solution to require an incremental approach over time, and we think this paper makes a significant advance in that direction.

2.      **Paper contribution:** is the general framework; the tool itself is presented only to demonstrate the application and viability of the framework. We would expect the framework to be implemented in more sophisticated tools in the future, perhaps embedded in modeling algorithms. Shortcomings in the tool, or simplifications for practical purposes, should not be equated to shortcomings in the framework.

3.      **Paper structure:** as the paper describes a large problem, reviews prior work and identifies gaps, presents a new framework to solve the problem, and successfully tests important portions of the framework with a proof-of-concept tool, leaving the rest to future work, we think the paper meets the requirements of a standard research paper.

4.      **Time:** the framework assumes a geomodel is assessed for consistency at a single point in time, that is, whether a configuration of geological objects (any physical geological entity, material or immaterial) at a timepoint is consistent with the rules of geology as captured by the hyperspace. The objects, of course, can have developed over different times, but it is their state at a specific time that is evaluated. This impacts the validity assigned to certain temporal relations, which might be invalid at a timepoint, but valid across timepoints. There are two main reasons for this choice: (1) practically, most models are developed to reflect a state of geological reality at a single timepoint (typically today); and (2) we expect assessment across time to reduce the number of invalidities, as many more situations are possible, dramatically increasing complexity and reducing the effectiveness of any approach.

5.      **Space-time:** the time assumption above implies the objects being assessed by the framework are so-called endurants or continuants, which are fully present at a timepoint, i.e. all parts that can be present at a timepoint are present, such as for a rock, geological unit, or fault surface. This contrasts with so-called perdurants or occurrants, e.g. space-time worms (4D spatio-temporal objects), processes, or events, which are not fully present at any timepoint, but unfold in time, so are composed of temporal parts that accumulate over time. Then only a temporal part can be fully present at a timepoint, but never the whole worm, process, or event. E.g. a ground shaking event - an earthquake - might have discrete early, middle, and late parts, but the whole shaking event is not fully present at any timepoint, because it requires all three parts to be complete. Our framework thus assesses endurants/continuants.

6.      **Material sharing:** we assume it is physically impossible for macroscopic (non-atomic) material objects (endurants/continuants) to share space at a single point in time, unless one is a part of the other. If models are evaluated at a single timepoint, then material sharing is impossible outside of whole-part situations (e.g. a lithology and a geological unit, or a formation and a member). We further assume material objects must be volumetric (occupy 3D space), and can share space with immaterial objects, either volumetric and non-volumetric. E.g. a filled hole can share space with its filling material, and non-volumetric surfaces and points on a material object do share lower-dimensional space with the object. Other non-material objects, such qualities, e.g. the colour, size, thickness, or shape of an object, also are deemed to share space with the object that carries them:  the grey colour of a rock is not made of material, but occupies the space of the material that carries it.

This is particularly relevant to folds: if folds are considered to be a feature derived from the shape of a material object, then folds occupy the space of the material object and can be seen as immaterial volumes. Therefore folds cannot have internal polarity, but might have a form of kinematic polarity, such as vergence and tectonic transport direction, which we do not address in this work, but could add in future work. E.g. in the future we might consider: if a fold

vergence contradicts the metamorphic polarity of a large orogenic terrane unit (90-180 degrees) then the situation could be inconsistent. Generally, folds will verge away from the core or deeper axis of an orogen and these directions might be useful in juxtaposition with other objects with polarity. We will adjust the text to better reflect these aspects.

Planar and linear fabrics are only loosely similar to folds in this respect: we consider them to be derived from a certain geometric arrangement of underlying materials and occupy the space of those materials. Then planar and linear fabrics are material objects that are part of their host rock, e.g. as grains or minerals or clasts, etc., arranged in a specific way. This allows fabrics to have internal polarity, as the fabric can gain or lose such materials over time. Given that we do not deal with parthood yet in this framework, and our work on fabrics is still on-going and in progress, fabrics will be considered more thoroughly in future work on parthood; but we believe the framework will accommodate them. Therefore, the material geo-objects we consider in the paper (depositional, metamorphic, intrusive, extrusive units, linear and planar fabrics) cannot share space at a single timepoint, but they can possibly share space with the immaterial objects at a timepoint (folds, fault surfaces, erosion surfaces). Overcoming this problem requires a solution for the parthood issue, which is future work. It is also tempting for simplification to consider material volumes that are tightly intermixed to share space, but this is a physical impossibility – these are simply objects with mixed composition.

7.     **Parthood:** as mentioned above, material wholes and their parts can share space at a timepoint (e.g. a lithology and a geological unit, or a formation and a member), but we do not incorporate this into the current framework, leaving it to future work. We expect parthood to be an additional dimension in our hyperspace, added to the space, time, polarity, and object type dimensions. However, we do not think its exclusion is a severe limitation: the exclusion does not invalidate the framework nor its use for the very many non-parthood situations; and parthood situations are not currently output by most modeling algorithms. All prevalent algorithms, that we are aware of, will partition material objects into non-overlapping regions, by design; so if geometric representations from these algorithms have spatially overlapping regions, then there exists an inconsistency. Consequently, we think the exclusion of material sharing is not a severe limitation for immediate practical purposes, but we do wish to overcome it in future work.

8.     **Metamorphic units:** considering the above points, there are no issues with metamorphic units sharing material with other units, because such units cannot share material at a timepoint unless one is part of the other. Material sharing is often invoked for convenience for metamorphic units, to allow earlier developed units (e.g. a protolith) to share space with one or more later developed units (e.g. metamorphic), but this is space-sharing (not material sharing) across time, and not at a single timepoint. As there cannot be material-sharing at a timepoint, there is no issue with our approach. The issue lies with the common invocation, which treats these objects as space-time worms, but that is a function of how geologists speak about such units, and not a limitation of our approach, which considers geological reality at a timepoint.

We recognize it is possible to have mixed compositions, in which at fine scales (possibly at the mineral unit cell level) some material is altered and adjacent material is not, but still neither material shares space with the other at a timepoint. Moreover, if the input data to a modelling algorithm contains distinct units for a protolith as well as for later metamorphic units, then virtually every current major 3D modelling algorithm (that we know) will avoid their overlap by design and generate non-overlapping partitioned regions for them. Therefore, the lack of material sharing is not a severe limitation for metamorphic units at present. In future work, it might be accommodated through the addition of the parthood dimension to the framework, if the different units are seen as temporal parts of a single material space-time worm.

9.     **Internal polarity:** in general, internal polarity in our paper refers to the direction of material accumulation or reduction. Note the issue of local vs global polarity is an implementation issue and NOT a framework issue. It simply concerns the selection of internal polarities used to evaluate consistency - local to an evaluation point, or global to the object. How the internal polarity is evaluated, and the nature of the polarity itself (material accumulation/reduction) does not change, thus the framework is unaffected. What is affected is the tool, which must coordinate local polarity data with evaluation points along an object, significantly complicating the tool. Indeed, because our tool evaluates all points on an object, we could easily swap in local polarities for the global polarities, at the cost of obtaining and managing the local polarities. To simplify our proof-of-concept tool, and because local internal polarity data was not available for our real-world case studies, we used global polarities: this is sufficient to test the framework. But such use of global polarity is not a defect in the framework, it is simply an implementation choice made to prove the overall concept. We will clarify this in the text and include more on local vs global polarity.

Some more notes on polarity: a move to local polarity in implementation, however, does offer some potential additional rewards, which could require some minor changes to the framework. After extensive discussion, we determined use of local polarity could identify some rare but incorrect geometric anomalies, i.e. errant local shapes for depositional units. This is offset by the potential to give rise to erroneous inconsistency results due to strange local effects: it is possible very local process behaviour could generate artificial inconsistent results at the local level, when they are valid globally.  Moreover, local polarities also are often difficult to obtain – e.g. not all implicit modeling packages expose the implicit vectors (which further are not always the same as our internal polarities), and field observations as well as other data are often sparse or missing. In these cases, global internal polarities are easier to estimate and use.

On the other hand, global polarity is sometimes hard to estimate, e.g. radial cooling directions for some intrusions, but even with these we think a vector from cooling center to an appropriate edge of the boundary will suffice in many cases. BUT, this is not an issue for the framework, because the framework does not use internal polarity (neither global or local) to determine consistency for pairs of objects involving intrusions or fault surfaces - only the age vector determines consistency for them. This is evident in the truth table involving these objects, where the consistency value is the same for both aligned and misaligned polarities. In essence, for some entities, polarity does not impact consistency; so the associated part of the

hyperspace is empty in those places. We will clarify this in the paper. We are also uncertain whether use of local polarities would require a minor modification to the polarity relation evaluation, e.g. by altering the angle threshold between vectors. Given these concerns, and need for more sophisticated tools and related data to test local polarities, we leave consideration of local polarity to future work.

**Review From Rob Harrap:**

I'll divide my comments into two sections. The first is changes I think should be made. The second, in the attached handwritten long-form discussion, are things the authors could consider to improve the manuscript but which are not essential. Note that the 'essentials' summarized here also occur in the handwritten notes, which also refer to page and line numbers.

Thank you for your extensive and informative comments. The proposed changes will be incorporated into the revision.

**Essential comments:**

P2, L16

Will expand to include human-led model/map generation.

P4 L4-7 Implicit knowledge in model correction.

Will clarify we target an explicit approach to identifying model inconsistencies, to promote reproducibility.

P4 L16

We understand that Harrap 2001 was also used for map consistency checking, and will point this out as unpublished work.

P4 L17: Range of knowledge. This could really benefit from an example to give explanatory context.

To be changed to: "Key goals for a consistency checker then include an expansion of the range of knowledge *to* include an enhanced representation of geological relations plus an approach for combining such relations into valid and invalid geological situations, for effective consistency evaluation."

Figure 1. I don't object to the figure, but I believe it needs a discussion in the body of how these errors happen. I realize that they are often 'black box application' errors, but was any work done to see if some are simply geometric? Can some be avoided by model construction

methods? A specific typology is not needed, but perhaps a few words here to give non-specialists a sense of why good tools produce errors?

These errors are driven by specific sparse data configurations. The key point is that good methods can create bad models under certain circumstances. Explaining why they create such models, beyond tying it to sparse data, is beyond the scope of this paper, as the technicalities are complex, and the issue is well recognized. We propose to add some words to this effect.

P4 L25

Noted.

P5

Thanks for the Gailleton reference. Note that de Kemp did related work using spatial agents in generative models. We intuit that models created by other generative processes could also benefit from a check against acceptable situations, as captured by the truth tables.

Re: purpose of a consistency-checking framework: we see our framework mainly as (1) a testing framework after a model is built, or (2) a testing framework for a model during its building process, which can influence its development  (your generative framework we think). Not sure about the link to legends - we see it rather as a representation of general geological knowledge, not any one legend or legends in general - legends are a subset of this knowledge.

P5 21-25. This is weak. This is well discussed in the literature. I'd either contract this to a sentence or do it justice by expanding it.

Will contract to a single sentence: "Geological knowledge has been accumulated over thousands of years of human inquiry into the natural environment, with modern formal geological knowledge emerging in the mid 1800's (Lyell, 1833; Rothery, 2016) culminating with several current formal ontological articulations (Refs. Perrin et al. 2011, Brodaric, B. and Gahegan 2006….).

P6 4,5 Geological 'area' - The term is vague. Domain? Area of common history? Think about reasonable geological cases where one would or would not put an 'area' boundary.

Propose to replace "area" with "situation": "Data then includes any form of observation used to understand a specific geological situation…"

P6 L6-26

Will make grammar corrections. We prefer to capitalize Truth Table, using it as a proper name for our artifacts, but will change to 'CC Truth Table(s)' so it resembles a proper designation.

P6, 18 I'd argue perhaps as an aside that some mapping is intended to identify contradictions or gaps in extant geological theory. In this case a contradiction is desirable in the short term.

Agreed. But the contradiction is still useful to identify, as not all contradictions need to be corrected. Modellers can choose to address them, or not.

22-26 There is a big conceptual jump here. 'These relations.' I think this really needs a bit of expansion and clarification as it is very important to your idea development.

Propose to change to:
"To determine knowledge consistency for a 3D geo-model, we expect local knowledge to be typically derived from a 2D map legend, cross-section, or associated report, with the geological processes and the combined event histories being discerned through geologically possible binary relations. For example, the contact relation between two adjacent depositional units can be decomposed into a spatial relation (spatially touching), a temporal relation (temporally adjacent), and polarity relations (aligned material accumulation), and each of these can be evaluated separately for consistency with established geological knowledge."

P7 13-17 Give an example of such a global/local disparity for context/clarity?

An example is the wide ranging variable polarity we get locally with a volcanic unit, but the overall volcanic stratigraphy is consistent (average vector) in a regional setting. We will add the example.

24 These vectors point roughly... not exactly. You eventually refer to them as normals. I think you need to make it clear that these are rough directions not precise vectors?

Agreed. Propose:

P.7 L.10 "The first type of polarity is an internal vector generally pointing in the direction of creation or destruction of an object's material. This is a generalized direction vector (local vector average) giving the trend of the object's polarity. E.g. for depositional units …"

27-28 not a volcanologist, but if you are in a vent of a volcano are the rocks extrusive or intrusive?

They are extrusive. Once subaerial or submarine exposure occurs, the result is a volume that is an extrusive unit.

P8 10-17  I find the discussion of metamorphism here tricky and perhaps even problematic. Your earlier vectors are grounded in locally observable phenomena. This might not be the case for metamorphism. I expand on this in the handwritten notes but perhaps the paper would be better by not worrying about the complexities introduced by metamorphism (other than to say

a unit can be identified as 'metamorphic.'). Like fold volumes, these ?superposed? 4d volumes are going to be messy...

Agree they can be messy, but we introduce metamorphic objects here to start the discussion since they are core to many complex terrains that have difficult to determine relations i.e. orogenic belts. We think a metamorphic gradient at a regional level, or in a more local sense adjacent to intrusions, can help with determining potential event history and model consistency. For example metamorphic gradients may indicate structural inversions, overall tectonic telescoping or magmatic sources important in mineral potential studies. Some consistency assessments are determined solely by age vectors and ignore the internal polarities (e.g. intrusive units). However, when the age gradient is not obvious, the internal gradients can play a role in determining what is more likely a valid geological relation. For other issues related to metamorphic units and material sharing see (8) above.

18 Some structural geologists would object and say they map fault volumes. Picky point but...

Agreed. Fault volumes have been distinguished from fault surfaces in geological ontology (Qu, et al, 2023). In this paper we do not consider fault volumes, only fault surfaces. Fault volumes are for future work. We expect them to have characteristics similar, but not identical, to fault surfaces, at the very least due to the different possible spatial relations (volumes vs surfaces).

Erosion surface - see in the handwritten my notes to ongoing work in the terrain simulation field on what erosion surfaces are. Probably out of scope for this paper but... an area of interesting parallel development in CS/graphics.

Agreed there are parallels, but more nuanced notions of erosion surfaces are out of scope for this paper. Also see (6) above for fold volumes.

P9 5, 9  Fabric versus elements. A fold axis is not necessarily a fabric???

We will clarify that fabrics are material objects, but folds are not and neither are fold axes. So folds and fold axes do not have internal polarity featuring material growth or reduction. However, they might have another kind of polarity, say kinematic, which captures the directionality of movement associated with related material objects.

15-16 In your intro you cite RMH and KLB a lot. I'd say KLB did a LOT of work thinking about fabrics and representation and you don't mention this in the intro. For example, Burns 1969, 1975 both discuss fabric chronologies...

Agreed, Kerry Burns made significant early contributions to fabric relations. We will expand the text to include this work.

20-21 This paragraph makes huge leaps. Perhaps 1-2s talking about topological relations like 'meets' so set up your examples?

Propose to modify to: "The patterns involve a chain of binary geological relations that have a specific spatial-temporal-polarity topology,  that taken together may add up to a complex geological situation."

P10 Fault has displacement because it is younger than bounding units.

Age relations between faults and other objects are definitely captured in the truth tables via the temporal relations.

P17 L16 Stratigraphic

Will remove "stratigraphic considerations" and replace with "following considerations".

P17 31 Cute math. Out of context here. Either expand or remove.

Will remove.

P18, Truth Table again. Not going to rant again 😊.

The term "Truth Table" comes from logic, in which the truth or falsity of a statement, including a relation, is specified. Given our tables specify the (in)consistency - "truth" - of complex geological relations / situations, they are in a loose sense also truth tables.  It might be more accurate to refer to such situations as possible models, as logicians might do, but we deem it too confusing to do so. We prefer to capitaize the terms and use them as a proper name for our artifacts. Perhaps we can say "CC Truth Table" as a proper name.

L 13 Metamorphic Units are non-abstract since they have material. Unlike fold volume.

Agreed. Metamorphic units are material objects. See (6) above.

P20, 20 Geodes – Solution is there a URL. Citation standards check.

Will will provide a URL.

P23 L8-10

Agree that a graph representation might accomplish the same. However, we would probably require a BREP to generate the graph. Note our tool is just proof-of-concept and we expect different tools and geometric representations to leverage the framework. The framework is not tied to any one tool or implementation, including our own; it stands apart as a general method.

Figure 13 - give approximate scale please

New figure with scale will be inserted.

P31 L13-14 I assume that this problem area is inside the volume of the model not near the boundary. If so, you can make the point that the chance of it being 'noticed' is very small, so such methods as you develop are essential. 'Looking' is not a practical alternative.

Thanks. Good suggestion. Propose to add: "This error would be difficult to detect through visual inspection alone and if missed could have profound effect on validity of downstream models such as a flow simulation."

P39 L1-2
My gut feeling … is that this may address situations of under-constraint: while very many models are possible, geological knowledge also suggests which are plausible. Dangerous ground to tread, though. Some people are working in that general area (Clair Bond in particular). Epistemology of field geology…

Thank you for the reference. We hope we also show that plausibility can vary amongst models with the same data, but different knowledge. The framework we propose will then allow for plausibility identification in the face of changing data density, types or configurations.

I'd like to see a short appendix (extension to your appendix) that provides some more info on development, tools etc. I realize this is probably available beyond the link but links have a way of disappearing.

We do not think more info on the tool is pertinent to the main point of the paper, which is the framework itself. As we view the tool as a throw-away, proof-of-concept, we think the current level of detail suffices, i.e. broad characteristics and notable limitations. A deeper look at the tool gets into implementation considerations that are out of scope in our opinion.

I do think that tightening up the generate versus test versus communicate angle would make your intro and conclusions stronger. In particular, while I 'get' your tables after a lot of time looking at them, they are not inherently useful as communication devices - a novice is going to struggle with them. This is tangential to your paper but in my view understanding is part of correcting.

We agree the tables are not intuitive, and take time and effort to understand. But the tabular format is the best communication device we found. We are open to suggestions for alternatives. In the end, our focus was on theory development in the paper, and proof-of-concept testing of a portion of the situations implied by the theory, rather than ways to efficiently implement the theory, which should follow.

Finally, probably a preprint issue but in your figures like 1, 2 I'd work to make the small blocks as large as possible for clarity. Some are quite small and you are not using the page.
Will expand the figures to use the full page.

**Review From Sam Thiele:**
This manuscript presents an interesting and relevant prototype approach for checking if 3D geological models are topologically consistent with fundamental geological principals. I see many applications of the proposed approach (if it is developed further and accessibly implemented), and suggest that it will be of interest to the readership of GMD. The overall approach is conceptually sound, though I suggest that several aspects should be better explained, or slightly adjusted. Hence, I recommend that the paper could be accepted after the following minor revisions:

Thank you for the informative and thoughtful review. Please see below for our proposed handling of your comments.

1. It is necessary to better define the concept of "material sharing" (perhaps a more appropriate term can be defined?) earlier in the manuscript, and discuss the resulting division of geobody types: those that cannot occupy the same space (e.g., intrusions, sedimentary units, etc.) and those that overprint and so co-exist with older geobodies (e.g., metamorphism, alteration, deformation). Also, as mentioned in a subsequent point, the important topological differences between "material sharing" and "volume occupying" objects should be specifically clarified.

Thank you for the feedback. We plan to explain this better according to the general comments above, especially (6).

2. I note that hydrothermal alteration is not mentioned in the types of geo-objects listed, and suggest that it needs to be included, given its general importance for many applications (e.g., mineral exploration). Either the "metamorphic" geo-object could be renamed to "metasomatic" (or a similar term) or, if necessary, a new geo-object type included.

   We think these are both metamorphic objects, at least in our ontology, and will improve our definitions accordingly. In effect, we anticipate no difference to the truth tables for these distinctions, as they represent variations on altering source material.

   Similarly, I suggest that "Fold volume" is a misleading term for an (apparently) broad geo-object class (e.g., I presume regional tilting, widespread shearing or other deformation, e.g., a fold and thrust belt, would be represented using this class?). Please consider renaming it to e.g., "deformation volume" or "tectonic domain".  .

These are different things to us. We will clarify in the text. A fold object is inherent in a rock volume with a particular generally differentiable continuous curvy-linear shape, and does not necessarily apply to all things created by some tectonic deformation process, especially if the underlying material is not characterized by the required shape(s). Nor does it include aggregations such as a fold or thrust belt, which is a collection of different features (both

material and immaterial), unless one discerns a fold in the underlying rock volume at a certain scale. See (6) above for more on folds.

3.  The Linear and Planar fabric geo-objects do not seem to fit well in this categorization, and are not discussed anywhere in the manuscript after being defined. I would suggest either expanding the explanation of these two geo-objects (and including them in the subsequent topological analyses), or defining them somehow as a different category of "geo-object" (i.e., as partially noted in the text, all of the previously defined geo-objects can be associated with linear and planar fabrics; hence fabric seems an inherent property of a geo-object rather than a distinct entity?).

    Fabrics are distinct entities in our ontology. They are derivative objects like folds following the GeoScience Ontology (Brodaric & RIchards 2021). In effect, fabrics in our ontology are derived from a certain arrangement of underlying materials, which are arranged in a linear or planar fashion. However, as explained in (6) above, because they are not fully tested and overlap with the parthood issue (they are seen as material objects that are part of their host rock) we will clarify they are included in ongoing and future work on parthood.

4.  While I understand the temptation of "completeness", consider defining / introducing (and presenting in your figures) only half of the topological relationships (Table 1 and 2), rather than including each relationship and its converse. As stated later in the manuscript, only one of each pair of (opposite) relations are needed: I.e., if "A contains B" is defined, then the "B is contained by A" relation becomes redundant. I would suggest either dropping these converse relations or presenting them clearly together (as for the temporal topology relations outlined in Table 2).

Not all converses are redundant as they are not symmetric. For example the 3D and 2D/1D spatial contains relations are not symmetric, so although a 3D volume can contain 2D surfaces and 1D lines, the surfaces and lines cannot contain the volume-. Likewise not all temporal converses have the same truth table value, so must be distinguished in the truth tables, thus should be fully explained in the ms. For full explanation, we believe the spatial and temporal relations in Tables 1 and 2 should be represented in their entirety. We are open to improvements to their presentation, but cannot conceive on how to improve them ourselves. Also, it is standard practice to include both the relation and its converse in presentation of these relations, both for reasons of theory completeness and to guide application and use. Note the asymmetry can extend to various geological relations, as not all geological relations will behave symmetrically wrt to consistency. Inspection of the truth tables will show that for certain object pairs the converse spatial and temporal relations do not have the same consistency value… the geological relation behaves asymmetrically. We will improve the text to better reflect these points.

5. It is unclear from the text (and from Table 1) how the 9-intersection (9I) model treats material sharing. E.g., If A contains B, do A and B share (some of) the same space, or not? Given the distinction between material-sharing and material-defining (for want of a better term) geo-objects, I think this is an interesting and important distinction in the geological context (which, if I understand correctly, the 9I model is fully capable of capturing, if properly explained). To clarify this distinction, a short example might be given in the text: e.g., if a sill (B) intrudes into host-rock (A), does A contain B? And what if some localized alteration (C) overprints host-rock A such that they locally occupy the same space (i.e., material sharing); does A also contain C? I suggest that these two geometries are topologically distinct in important ways, but that Table 1 does not make it clear which relation is appropriate in each case.

The 9I is agnostic to material sharing. In effect, the 9I spatial relations are conceived as occurring in abstract mathematical space and not in physical space. Extension into physical space requires further assumptions and related limitations about the ontological nature of the underlying objects, which is not present in the original 9I work. In related work, we have shown how this can be done for the containment relation (https://link.springer.com/chapter/10.1007/978-3-319-01790-7_22). In this work, we strive to do something similar for geological relations, to build the hyperspace of possible geological relations / situations. Valid geological relations thus are best understood through the truth tables. We will clarify the distinction between material-sharing and space-sharing as outlined in the general comments (4)-(6) to provide more clarity on this issue.

6. The spatial relationships associated with faults and unconformities should also be explicitly addressed, as this has big implications for the subsequent model validation approach. While I understand the need to model these as 2D (embedded) surfaces, I suggest that (for the needs of this paper) they are better treated as arbitrarily thin volumes. E.g., If older and overturned sedimentary unit A is juxtaposed against a non-overturned sedimentary unit B across fault *F*, is there an *A meets B* relation? Or just *A meets F* and *B meets F* (such that the fault is treated as an arbitrarily thin volume). In the former case, the *A meets B* relation would lead to an incorrect invalidation of this geological model following the Truth Table (Table 3), while in the latter this situation is elegantly avoided. Equivalent arguments could also be applied to unconformities.

We will clarify the spatial and material nature of all objects, as outlined in the general comments above. In our ontology, faults surfaces are by definition not volumes and are immaterial. Fault volumes are a different entity, being volumetric and material, and are not considered in this paper and left to future work. Our truth tables reflect the fact that touching units with certain age relations will be inconsistent and this can signal the need to include an intermediary object, possibly a fault. I.e. the polarity relation evaluating the internal and age polarities of A and B would return unaligned, which combined with the meet relation would return inconsistent from the truth table. Now, you are likely raising the practical problem of

how to determine the spatial relations between adjacent volumes split by an intermediary surface, because the volumes appear to touch each other as well as the surface. We suggest this is an implementation problem, and not a problem with our framework: it involves excluding a spatial meets relation from the spatial matrix e.g. between A and B, because of the intervening fault, and including A spatially meets F and F spatially meets B. Our tool in effect does this pre-processing based on the types of objects, aligned with your intuition. Without the pre-processing, and the exclusion of the A (spatially) meets B relation, an inconsistency would be raised for this situation, which would also be a reasonable result we claim. However, we do not see how implementing an arbitrarily thin volume would help: A would still touch B because the fault volume has no thickness; further pre-processing of the model to add thickness to faults, thus changing model geometry, seems too risky as it would possibly lead to other erroneous results. We think we should retain the original spatial (and temporal) integrity of the model, and accept the inconsistency or perform pre-processing to filter (in)valid spatial relations based on the object types. Note some pre-processing is likely necessary regardless of geometric paradigm: e.g. extraction of spatial relations directly from the scalar field of an implicit model would require spatial pre-processing almost certainly having some knowledge about the ontological nature of the objects. Perhaps there are other, better, solutions to this problem.

7.  While the limitations of representing polarity as a single vector are correctly acknowledged, I suggest that this could be done so earlier in the manuscript. Furthermore, it might be worth mentioning that, given tendency is a vector field (often equal to the gradient of a scalar field used for interpolations), it could be better defined at every voxel or mesh element in a model. If implemented like this, alignment or opposition could be defined at every discrete location in the model, along with the local topology (i.e. based on the adjacent voxels / mesh elements), and the truth-table applied to identify local geological consistency or inconsistency. A model would then be considered globally consistent if all its parts are locally consistent, and any inconsistencies could be quickly located.

Agreed. But this is left to future work. It is in fact an implementation issue: as you correctly point out, the framework is agnostic about the global vs local nature of the internal polarity, it is just the tool that is sensitive to it. See (9) above for more on this.

Other minor comments:

Page 1, Line 17: What is "natural storage"? I suggest "geological storage" or just "storage".

Agreed. WIll modify.

Page 1, Line 19: "as well as minimize user problems" – this is a very vague statement. Can the authors be more precise?

Agreed. We will remove this statement, as it is not critical to the abstract.

Page 1, Line 22: Consistency with what? Please clarify.

Agreed. Will modify to "consistency with general geological knowledge"

Page 1, Line 23: "Space of consistent and inconsistent geological situations" – does this approach define a "space"? I suggest instead that the truth-table is a set of rules. If "space", then what are the dimensions?

Yes it defines a space of (im)possible geological relations, with space, time, polarity, and object type axes. See (1) above.

Page 2, Line 1-8: Implicit interpolator quirks (e.g., bubbles) should also be mentioned here, as they seem like a likely source of inconsistencies.

Agreed. Will do.

Page 2, Line 8 (and 10): Why equiprobable? Surely a set of model realizations will contain many probable models, as well as some less probable ones (that are still consistent with the data)? I.e. samples drawn from a posterior distribution won't be equiprobable?

They are all equiprobable if we have a) no priors, b) they are drawn from the same data, or the data is randomly perturbed. Agree that intrinsically the models will have greater or less probability, but how do we know which it is? That is what the CC is for, to push some into the less probable direction when they are not consistent. What is left is at least more probable than those we reject. This impacts on the discussion of geological reasonableness. We will clarify equiprobability in the text.

Page 2, Line 12: Lyell, 1833 and 2022 – that's an impressive career!

Thanks. Will fix.

Page 2, Line 15: Clarify what is meant by "hypothesis testing"

Will clarify.

Page 2, Line 18: I would suggest that the number of models is not an issue; their quality (and validity) is. Rephrase.

Agreed. Will modify to focus on quality.

Figure 1f: How is the topography relevant? I would remove this.

The topography indicates how the data biasing occurred BUT this topography needs to be checked. Will revise if necessary..

Page 4, Line 4: Note also that the visualization of complex geomodels is a significant challenge, making manual validation difficult, time-consuming and likely to miss problems.

Agreed. Will add text for that.

Page 5, Line 1: Important why? Please explain.

For computer encoding of allowable object relations consistent with, for example real world physics. Will modify.

Page 6, Line 3: Is topology (e.g., unit A intrudes unit B, foliation C crosscuts foliation D) knowledge or data under this definition? I would suggest that the observed relation is data, but its implications for timing is knowledge. Given the importance of spatial topology for this work, and its link to timing, I would suggest clarifying this in the text (perhaps with a small example).

Agreed. Will modify.

Page 7, Line 19: "age direction vector" – I would suggest that this is a (directed) temporal topology relation, not a vector.

Yes, it is a directed temporal relation, but such relations have no direction in space. For polarity evaluation purposes it is useful to treat it as a vector in space to enable comparison of directions with the other (internal polarity) vectors, to determine polarity (mis)alignment. In essence, the temporal relation is not directional in space, it is directional abstractly, and it is convenient to orient it in space for polarity relation assessment. We will clarify in the text.

Page 15, Line 2: Use appropriate term from Table 1 (meets?)

Agreed. Will do.

Page 17, Line 23: "covers" and "covering" – distinguish these from the "cover" relationship defined in Table 1?

Agreed. Will replace with 'overlies' and 'overlying'.

Page 19, Line 3: Material sharing has not yet been defined?

We will move some general assumptions to the start of Section 2, including material sharing.

Page 19, Line 8: "Relata" and "relatum"; why not "relations" and "relation"?

A relation is an interaction between participants. The participants are distinguished from the interaction itself: the collection of participants are the relata,  and each individual participant is a relatum. Will clarify in the text.

Figure 4: What is BREP?

Common 3D worker usage: Boundary REPresentation for objects in 3D models. Wil provide a specific reference from Braid, Ian C. "The Synthesis of Solids Bounded by Many Faces." Communications of the ACM 18, no. 4 (1975): 209-216.

Page 22, Line 15: Note the strong link between spatial and temporal topology (e.g., xcutting relationships etc.; Thiele et al. 2016, many publications of K. Burns, etc. )

Noted.

Page 23, Line 5: This would arguably be much simpler / easier in a discretized (gridded) form? As topologies could be rapidly computed for cell neighborhoods and then aggregated.

In some ways yes, perhaps faster, but then there would be sacrifices with using a specific cell resolution. An earlier paper (Schetselaar and de  Kemp 2006) used a voxel representation but did have issues with loss or under-representation of volumetric features. A BREP can represent better the non-material lower dimensional features, fault surfaces, folds, fabrics. Lots of implementation issues here but we give a start with BREPS because we think it's the way forward for a wider range of geological features. Note this representation is required for our specific implementation, and may not be required by others.

Page 23, Line 15-21: Consider using the "structural topology" and "lithological topology" terms from Thiele et. al., (2016) to more clearly explain this distinction.

We plan to add "'Other mechanisms may also be appropriate for representing multiple relations for parts of objects as in Theille (2016) such as with stratigraphic and structural network graphs."

Page 24, Line 1: Can the "Equivalent to" temporal relationship be used to resolve this issue, by building a graph of the more detailed structural topology and expanding the temporal relationships to fit this by adding "equivalent to" edges between all structurally distinct volumes of the same lithology?

Agree, that is one approach. Another, perhaps simpler, approach would have the spatial and temporal matrices built using each distinct unit fragment rather than the unit as a whole. Then no change is needed to the tool. Again, this is an issue of implementation, not concept. We decided to use the whole to simplify presentation in the paper, allowing the spatial and temporal matrices to be presented in full. The tool could have worked equally well if each fragment was a distinct object in the matrix.

Figure 14: What is the y-axis in this figure?

The Y–axis has no properties. It is used to help readability and interpret readability if stratigraphic or structural temporal overlaps were present.

Page 38, Line 19: What makes a model "ideal"?

Good point. Will clarify an ideal model as "one which is close to reality and matches the constraint data".

Page 39, Line 19: What is "parthood"?

Parthood is a relation between a whole and its parts. It is used here to signify the possible inclusion of whole-part constraints in consistency checking. The term will be defined in the general assumptions.

Further minor language suggestions are included in the attached pdf.

Thank you. Will modify as required.

**Review From Michel Perrin:**

**INTENTION OF THE PAPER**

The paper addresses an issue of particular importance: checking the validity of multiple equiprobable geo-models possibly constructed by using automated methods. It is a pioneer work that examines how the application of formalized geological rules can help deciding which models are valid and which are not.

In order to let properly appreciate the interest of the paper and its possible limitations or defects, I will first operate a critical analysis of the text. I will then formulate a general advice and some research suggestions.

Thanks for the useful feedback and insights, which will improve the paper. We do, however, strongly disagree with several of your points and provide a rebuttal below.

**TEXT CRITICAL ANALYSIS**

**Part 2 Geological Consistency Checking Framework**

**Considered geological entities ("geo-objects")**

The term "geo-object "is confusing because these entities have different natures:

"Geo-object" is used in our paper as a general term for any geological entity of interest. For the purposes of the paper a geo-object denotes an instance of a type defined as a collection (disjunction) of certain geological entity types of interest (depositional unit type or intrusion unit type or …) Instances of this collection can have different natures without impacting the collection itself. This collection is limited to the paper and not to general geo-ontology.

**Polarities**

- The **gross polarity between two "geo-objects"** is merely an age relationship (older to younger)

  There is a difference: the relation itself has no spatial direction (it may have an abstract direction with a converse, if not symmetric), but the polarity is a vector with a spatial direction, which makes the evaluation of the polarities easier, based on vector angles and facilitates the comparison with internal polarity properties that are spatial either globally or locally.

- The **internal gross polarity** with a geo-object is a growth direction vector which indicates the direction in which the geological process associated with the geo-object has progressed. Geologically this makes sense for most of the considered geological

entities with some exceptions (faults, not depositional planar fabrics). In some cases, the internal gross polarity may be difficult to characterize. This is the case in  an  intrusive in which  local polarities correspond to the cooling directions. The gross polarity is related to the field of local polarities in a way that the authors do not specify.

Agreed that gross polarity is difficult to estimate in some cases. But we think these are somewhat limited and much can be done with approximations. E.g. even in intrusive radial cooling situations, the geometry of the rock body can often allow for a representative vector from a center. However, it is important to note that as discussed in (9) above, the use of gross polarity is an implementation issue, a simplification for our proof-of-concept, and does not impact the nature of the framework overall nor its testing. We discussed this issue at length and are reserving exploration of local polarities for future work and think the framework works for them also.

The authors specify how gross internal polarities can be used in practice only in the cases of depositional units or intrusive. In the cases of metamorphic units, folds and not depositional fabrics, the interest of their use is not demonstrated.

The purpose of the paper, as outlined in (1)-(3) above, is to introduce the framework and test enough of it to provide grounds for evaluating its overall merit. As the problem is very large, we can only develop and test a portion of the framework. To this end we have specified and successfully tested combinations of Depositional Units, Intrusion units, and Fault and Erosion surfaces. The remainder is for future work.

**Spatial relations**

They are considered only in the case of "spatial objects" i.e. 3D /2D / 1D entities (geological volumes / geological surfaces / lineations).

Abstract spatial relations in qualitative spatial calculi are developed only for abstract spatial objects. Their application to physical space requires additional restrictions (e.g. for containment see https://link.springer.com/chapter/10.1007/978-3-319-01790-7_22). Table 1 therefore begins to add these restrictions, but additional considerations need to be factored in such as material sharing, which is provided by the truth tables.

**Truth tables**

- Some of the general principles exposed in page 17 suffer exceptions:
- *Lateral continuity*: heterochronous geological units are not uncommon.

  Agreed. This will be fixed. Our framework is not dependent on the poor wording in our original definition.

- *Paleontological identity*: geological units are homochromous only if they bear the same association of stratigraphic fossils.

  Agreed, but also fossil temporal correlation is also subject to large error ranges since species can evolve over large (human time frame) time ranges. Again, this does not impact our framework itself, but might impact the nature of the input temporal information. We will modify the text.

 *Principle of inclusion*: not applicable to geodes for instance.

 Agreed. We propose to reword to include mechanically incorporated entities, such as clasts in a conglomerate or a volcanic flow picking up older material.

- The authors mention that 45 truth tables can be established corresponding to "45 valid pairwise combinations of objects" but the example they give in table 3 only corresponds to the simplest of these combinations: the combination of two depositional units. I am not convinced that they could illustrate as well any of the 44 remaining cases.

We provide 6 truth tables for combinations of Depositional units, Intrusion Units, and fault and erosional surfaces, and test these in several case studies. However, we have preliminarily developed and tested the remaining tables and are confident the framework works for them. However, each table requires very lengthy and careful consideration and we have not yet been able to complete this. Coming to a final state on these tables requires a larger effort that is beyond our reach, and this paper, at the moment.

- There is a punctual mistake in table 3. The case corresponding to the cell of column 1, line 3: Depositional Unit A spatially meeting Depositional Unit B but temporally preceding Unit B with aligned polarities for the two units, is valid. The temporal interval of time between the depositions of units A and B may simply correspond to a period of non-deposition

We assume a period of non-deposition results in an unconformable surface. Therefore, the two units cannot touch directly, but each must touch the surface. If they do meet, then the surface is missing, which contradicts our completeness assumption, therefore is invalid. Reviewer 2 brought up the question of how to practically address the touching of volumes with intervening surfaces (see above), which is a practical issue that can be resolved in different ways.

- **Material sharing**: the material sharing rules adopted introduce **severe limitations**. For instance, the space-time association between a stratigraphic unit U1 of rank 1 and a unit U2 of rank 2 part of U1, will be considered invalid.

We disagree these limitations are severe, and hold that they can be progressively added in future work without invalidating the current framework. See (6) above. We plan to expand the text on this in the paper.

- **Spatial relation matrix**: the authors warn that "the entities related are the whole objects" and not fragments of them. This simplification is operated for practical computation reasons but the authors admit that it is problematic. They indicate that, in the case of an object A having multiple spatial relations with an object B, "the most dominant relation is selected". But they do not specify how this selection is operated. This introduces another **severe limitation** in their method.

We disagree this is a severe limitation. Firstly, it is a simplification made for the presentation purposes, and is not a limitation of the tool nor the framework. In fact, if we choose to include object fragments rather than wholes in the matrices, both the tool and framework would operate appropriately. The choice to use wholes was made simply to be able to ease illustrations in the paper. We could have equally chosen to represent the fragments in the matrices and diagrams, at the cost of bigger and more complex illustrations that are harder to comprehend because of their size. Secondly, the process of selection for the case studies was done by manually choosing the most frequent spatial relation after construction of a spatial matrix using the fragments. This was deemed sufficient for our case studies as most case studies did not have this problem of entities having multiple spatial relations with each other; in the one case-study where it occurred, it was very minor and easily resolved manually.

**Part 5 Discussion**

The point about **spatial relations** mentioned in page 37 lines 4-7 is of special importance. In my opinion, it may be easier to consider surface rather than volume representations as relata in the special relations considering that:

We disagree. Surface representations are not sufficient for consistency-checking. Volumes, including geological, have properties that surfaces don't have. For example, future work will address the parthood issue, but parthood relations such as between a group and formation, or a unit and its material, are much more naturally represented volumetrically and would be very complex to represent with surfaces. We think that consistency-checking should deal with the properties of objects in reality, and not of properties of their representations, such as a volume represented as a surface.

- geological surfaces belong to two categories: polarized (horizons, erosion surfaces, intrusive external boundaries) and not polarized (faults, thrust surfaces),

- there exist only three types of relations between two geological surfaces: disjoint, interrupts /interrupted_by , splits /split_by,

  Note the spatial relations in Table 1 are not geological relations - they are only spatial relations - and we are confident they are correctly depicted. Valid geological relations are captured by the truth tables in our approach, which reflect the allowable spatial,

temporal, and polarity relations between objects. However, we disagree the spatial relations you note above are complete for surfaces:  e.g. a fault surface could be part of a larger fault surface (or system of fault surfaces), thus they will exactly spatially overlap in places. An late extensional reactivation of a regional thrust for example.

- a majority of geo-modelers rest on BREP representations.

We disagree. Most state-of-the-art packages these days use Implicit modeling techniques, which have an internal representation of  scalar fields. The fields can then have various external representations such as BREPs, surfaces, voxels, etc.

**GENERAL APPRECIATION AND DISCUSSION**

As I mentioned before, the paper addresses an issue of paramount importance. In its principle, the validation approach presented is interesting but it is extremely ambitious. The authors intend to establish general validity rules resting on only three kinds of relations (spatial, temporal and polarity relations) and to apply them to a large variety of geological entities: geological units (objects), erosion surfaces, faults (surfaces), metamorphic assemblage, folds, fabrics (geological structures). The use of validation tables helps making this approach operational.

I see some difficulties in such a general approach. **Internal polarities** are defined as the directions in which some geological processes develop. This applies to sedimentation, erosion, magmatic intrusions and extrusions, contact and regional metamorphism and folding. However, the authors only consider the practical use of internal polarities in the case of sediment deposition. I see two difficulties in using internal polarities in the other cases for several reasons:

- In the cases of magmatic intrusions or contact metamorphism local internal polarities, are elements of a gradient. Defining a gross internal polarity, in these cases, may be problematic.

We agree this can be the case for certain (but not all) rock geometries. However, this is not an issue for the framework, because polarities are not used to determine consistency for relations involving intrusions or fault surfaces - these parts of the hyperspace are empty. In (9) we discuss the possibility of using local polarities for various scenarios in future research.

- In some other cases like not depositional 3D fabrics, no internal polarity can be defined.

Work on fabrics is in progress, but not in scope for this paper;  see (6) above. We have developed preliminary truth tables for them, but have not yet fully tested them. Generally, we

hold that fabrics are material wholes composed of smaller material parts arranged in a certain way. As a material object they can have internal polarity.

- The authors give no examples on how gross internal polarities could be used in cases other that those related to sediment deposition. I hardly imagine for instance how fold vergences could be used as a validity criterion.

Like fabrics, folds are part of work in progress; also see (6) above. We have developed preliminary truth tables for them, but have not yet fully tested them. Unlike fabrics, folds are immaterial objects that do not have internal polarity, so such polarity is not a criterion for their validation. However, other types of polarity, e.g. related to kinematics, might be useful to add in the future.

Another difficulty is due to the fact that the authors consider together 1D, 2D and 3D entities and put a special attention on geological volumes. As mentioned above, this put severe limitations to their approach due to:

1/ the issue of material sharing,

We disagree this is a severe limitation, and explain why in (6) above.

2/ the fact that only whole units are considered and not their partitions in entities of higher stratigraphic rank or their splitting into geological blocks due to faulting.

We disagree this is a severe limitation, and explain why in (7) above.

The authors claim that their paper presents a proof of concepts. But the concepts that they actually prove, cover but a small part of the ambitious issue that they address. The paper then consists for a good part in a list of interesting but yet unsolved research issues. For these reasons, it doesn't fully meet, in my eyes, the requirements of a classical research paper.

We disagree this is a severe limitation. We propose a general framework, and demonstrate that a portion of it works as expected. Refinement and testing of the rest is left to future (and ongoing) work.

Furthermore: we think that not all problems are solved in one paper, but require an incremental approach, especially large problems. We believe we meet the standards of a classical research paper as we outline a large problem, discuss related work and identify gaps, provide a general solution, and successfully test the solution on a part of the problem. As such, the basic elements of a research paper are met.

**RESEARCH SUGGESTIONS**

Some additional research efforts may be necessary for presenting a fully convincing research paper. Let me tentatively suggest two research issues that could be studied in priority.

1/ As I mentioned above, a solution for overcoming the difficulties related to material sharing and unit partition, would consist in basing validity checking mostly on geological surfaces, whose spatial, temporal and possibly polarity relationships can easily be characterized.

Thank you for the suggestions. As argued above, we believe the material-sharing problem is not a significant issue outside of parthood, and we are experimenting with various solutions to overcome the parthood problem.

2/ Fully investigating the case of regional metamorphism could be an interesting approach for demonstrating the applicability of the proposed method in a case other than sediment deposition. In this case, polarities are aligned with the temperature/pressure trajectories. Polarity inversion or gaps in the metamorphic facies succession will signal tectonic discontinuities.

Agreed. A focus on objects beyond the four tested in this paper is definitely a focus in  ongoing and future work.

Conversely, I suggest to simply ignore, for the time being, the cases of geological entities like folds and not sedimentary 3D fabrics. These deserve a deep geological analysis and might be object of deep investigation in a second stage.

We agree that a deeper test of folds and fabrics is part of future work.

**MINOR REMARKS**

Some figures represent geology in an unusual way and possibly confusing way.

- **Figure 2**

**2b, 2c:** the "roots" of the extrusion/ intrusion are not represented,

Thanks. Will fix.

**2d**: metamorphism generally decreases from bottom to top; in the figure the heat source to be "suspended" between bottom and top,

Will modify, keeping in mind it's a contact aureol of an intrusion not a regional geothermal gradient.

**2i**: doesn't represent a fabric but a single surface.

Thanks, will fix..

- **Figure 3:**

In **case a** (up left), the stratigraphic succession is represented in a reverse position (correct but unusual)

It is common in certain geological environments, such as the Canadian Shield.

- **Figure 9** is difficult to understand and seems to be in contradiction with the event history presented in figure 10:

We do not see any contradiction between Figures 9 and 10: each object is temporally preceded by (code 13) other objects.

- Horizon B is supposed to be anterior to the two faults but seems not to be split by these faults

Yes, that is true, but is simply a matter of illustration scale. Horizon B is cut but not 'significantly' offset by the faults in the model.

- The intrusive unit is represented in a funny way (external surface or volume?); it seems to stop on the green fault while it is supposed to be posterior to this fault.

Yes that is the case. The intrusion is a material volume with a spatial meets relation with one of the faults. This is possible, but not common in nature.

- **Figure 13**; the pattern transparencies make the figure confusing and do not help understanding the details of the geology.

Thanks. We will strive to improve the figure.

References to add:

[1] Garcia, L.F., Abel, M., Perrin, M., dos Santos Alvarenga, R., 2020. The GeoCore ontology: a core ontology for general use in geology. Comput. Geosci. 135, 104387.

---

## Referee Report (RR1)

Michel Perrin

Review of the paper Egusphere-2024-1326
Consistency-Checking 3D Geological Models
Marion N. Parquer, Eric A. de Kemp, Boyan Brodaric, and Michael J. Hillier`
(Revised Version)

This revised version of the paper « Consistency-Checking 3D Geological Models » is similar to the former version in its broad lines. However, it provides detailed additional explanations about the nature of the « geo-objects » considered for geo-model evaluation, about their internal polarities and about the space, time and polarity relationships that are of interest for consistency checking. Echoing many remarks and questions of the reviewers of the former version (including myself), this new version also corrects several defects or ambiguities of the former text and better explains some of the choices made by the reviewers at the implementation level. Together with the extensive answer that the authors addressed to the reviewers, this new version allows a better understanding of the paper intention. In view of this, I will newly examine some key points of the paper.

1/ The nine "geo-objects" that are considered for consistency evaluation, were selected in view of their frequent presence within geo-models. They can be 3D material objects (geological units), 3D "immaterial objects" (folds) or immaterial objects of lower dimension (erosion or faut surfaces, lineations…). Age and space relationships are commonly considered between geological entities of different natures but the use of a parameter like "polarity" is less evident. The authors provide examples illustrating how "polarities" can be attached to the nine categories of geo-objects that they consider. But they do not provide a unique clear definition of polarity that would be valid in all these cases. For this reason, it had some difficulty in considering that a unique internal polarity parameter could be attached to these heterogeneous objects in order to define a general validation framework. However, when considering the extensive explanations and examples given by the authors, it appears that they call "internal polarity" a vector which indicates the direction in which a geological process has been progressing through time. This direction may be identified in material or immaterial geological objects (sedimentary stack, metamorphic isogrades, folds) or deduced from the very nature of some of these objects (erosion surface). Internal polarities can provide information about the courses of a majority of geological processes: earth material creation, mechanical erosion, matter transformation (metamorphism), geological volume deformation and displacement. This includes folding but possibly also faulting and thrusting since displacement vectors can be associated to the geological volumes separated by a fault or a thrust surface.

Geological surfaces like horizons or faults characterize key instants of the geological history of a given region or site. For this reason, they are considered essential for capturing the essence of geological events (Wellmann & Caumon, 2018). Accordingly, in the classical vision, geo-model consistency mainly relies on a correct interpretation of the age and topological relationships between geological surfaces. By introducing internal polarity as a third parameter, the authors notably extend the spatial and temporal fields available for geo-model evaluation. In the spatial dimension, not only geological surfaces but also volumes can be

objects of interest since their local internal structure (stratification, sequential layering, basalt flow fracturing pattern successions etc.) can be used for determining local polarities. In the temporal space, this parameter can help tracing geological evolution continuously and not only at discrete instants. The evaluation methodology involving polarities that is proposed in the paper, can probably be applied to most of the geological entities presented in geo-models, including faults as long as they are not seen as isolated discontinuities but as parts of a dynamic system. In view of this, we may consider that the authors define indeed a novel interpretative framework.

It remains that, even in its new version, the text of the paper doesn't allow the reader to fully catch the full geological dimension of this novel approach. Erosion surfaces, individual elements of planar fabrics like metamorphic isogrades or sedimentary layer interfaces are all linked to some geologic process. An erosion surface materializes the ultimate stage of a rock matter destruction process, metamorphic isogrades materialize stages of rock volume burying process at depth, the ChronoBottom, the layer interfaces and the ChronoTop surfaces of a sedimentary unit materialize the beginning, the intermediate stages and the termination of a deposition process. All this would have deserved to be clearly exposed as well as the links which exist between the internal polarities attached to geological volumes, to geological surfaces and to fabrics, and also with the temporal polarities defined between meeting "geo-objects". All these parameters are related to a consistent geological whole. The paper redaction would be greatly improved if all this was exposed in a few lines together with a clearcut definition of polarity. This would help the reader realizing that what the author define is indeed a novel conceptual framework.

2/ The authors have made the risky choice of presenting in a single paper the broad lines of their geo-model evaluation method both at the theoretical and at the practical level. This choice is challenging for several reasons.

At the theoretical level, you must clearly define the polarity parameter which is at the heart of the proposed method at the geological level and also at the ontological level, since polarity is related both to material and immaterial "objects". At the ontological level, you should make a choice between various possible reference frames like GeoCore or GSO and give enough explanations about these tools to allow full understanding by readers who are not all familiar with the ontological reference that you have chosen.

At the implementation level, you must carefully choose the parts of the method that you will consider. The presented implementation should be kept simple in order to be easily understood but large enough for fully illustrating your approach.  Simplifications are unescapable but you should carefully justify each of them for not being accused of oversimplifying the system.

Use cases have to be chosen for fully illustrating how the methodology can be used in practice. The authors have taken the risk of not only presenting synthetic cases but also real ones based on results provided by industrial geo-modelers currently in use. This is another challenge since it is not easy to identify significant industrial examples.

In its present version, the paper tries to fulfill a majority of these requirements.  The careful analysis that I made of the theoretical background of the method shows that what is exposed still doesn't cover all the aspects of the subject but this can hardly be the case in a paper that intends to present all the broad aspects of a novel methodology both at the theoretical and at

the practical level. The comments made by the of the former reviewers (including some "severe" critics that I made) and the discussion that followed, contributed, I think, to produce a better text. Any new adjustment operated by the authors to still improve this new version will probably be welcome. We could also go on having ping pong exchanges on such or such details of text with the hope of making the text optimal but the added value of such exchanges might be little compared with the interest of quickly presenting a novel approach that defines a new interesting framework for geo-model evaluation. In view of this, I consider, this new version is eligible even as it is, for a publication by the Geosphere Special issue: The Loop 3D stochastic geological modelling platform – development and application.